# Multifunctional metal-organic framework-based nanoreactor for starvation/oxidation improved indoleamine 2,3-dioxygenase-blockade tumor immunotherapy

Liangliang Dai 📵 [1,2], Mengjiao Yao[3], Zhenxiang Fu[1], Xiang Li[3], Xinmin Zheng[3], Siyu Meng[1], Zhang Yuan[1], Kaiyong Cai[4 ✉], Hui Yang[3 ✉] & Yanli Zhao 📵 [2 ✉]

Inhibited immune response and low levels of delivery restrict starvation cancer therapy efficacy. Here, we report on the co-delivery of glucose oxidase (GOx) and indoleamine 2,3-dioxygenase (IDO) inhibitor 1-methyltryptophan using a metal-organic framework (MOF)-based nanoreactor, showing an amplified release for tumor starvation/oxidation immunotherapy. The nanosystem significantly overcomes the biobarriers associated with tumor penetration and improves the cargo bioavailability owing to the weakly acidic tumor microenvironment-activated charge reversal and size reduction strategy. The nanosystem rapidly disassembles and releases cargoes in response to the intracellular reactive oxygen species (ROS). GOx competitively consumes glucose and generates ROS, further inducing the self-amplifiable MOF disassembly and drug release. The starvation/oxidation combined IDO-blockade immunotherapy not only strengthens the immune response and stimulates the immune memory through the GOx-activated tumor starvation and recruitment of effector T cells, but also effectively relieves the immune tolerance by IDO blocking, remarkably inhibiting the tumor growth and metastasis in vivo.

[1] Institute of Medical Research, Northwestern Polytechnical University, Xi'an 710072, PR China. [2] Division of Chemistry and Biological Chemistry, School of Physical and Mathematical Sciences, Nanyang Technological University, 21 Nanyang Link, Singapore 637371, Singapore. [3] School of Life Sciences, Northwestern Polytechnical University, Xi'an 710072, PR China. [4] Key Laboratory of Biorheological Science and Technology, Ministry of Education College of Bioengineering, Chongqing University, Chongqing 400044, PR China. ✉email: kaiyong_cai@cqu.edu.cn; kittyyh@nwpu.edu.cn; zhaoyanli@ntu.edu.sg

The starvation therapy represented by glucose oxidase (GOx) has been recognized as a "green" strategy against cancer, as it cuts off the necessary nutrient supply to tumors with negligible side effects[1]. GOx has been applied for cancer starvation therapy, due to its ability to exhibit the immunostimulatory effects[2–4]. It can effectively kill tumors by emulatively depleting glucose and generating cytotoxic reactive oxygen species (ROS), thus, facilitating the exposure of tumor-associated antigens (TAAs) for an overall antitumor effect[5]. However, its immunostimulatory effect is naturally inhibited owing to various negative feedback immune resistance mechanisms[6]. The blockade of negative regulatory pathways combined with starvation/oxidation therapy represents one of the most promising strategies for tumor immuotherapy[7].

The immune checkpoint protein indoleamine 2,3-dioxygenase (IDO) is highly expressed in tumors, which can inhibit the effector T cell proliferation and induce the expansion of the T regulatory (Treg) cells by catalyzing tryptophan (Trp) to kynurenine (Kyn), thereby presenting itself as an attractive immune-therapeutic target for relieving immunosuppressive microenvironment[8,9]. The recent studies suggested that the IDO-specific competitive inhibitor, i.e., 1-methyltryptophan (1-MT), could effectively relieve immune evasion[10–12]. However, a modest anticancer immunity was demonstrated for the IDO-blockade monotherapy due to insufficient antigen presentation and immune response[13]. Thus, the combination of the 1-MT-mediated IDO blockade immunotherapy and GOx-activated starvation/oxidation can be a feasible strategy against tumors with strong immune response and weak immune resistance.

As the poor bioavailability and rapid inactivation of GOx and sequential biological barriers result in limited tumor penetration and low endocytosis, the construction of multifunctional nanosystems for efficient transfer of GOx and 1-MT in tumors is crucial to enhance the therapy efficacy[14,15]. On one hand, the metal-organic framework (MOF)-based nanoreactors combining the advantages of MOFs (e.g., high loading capacity and good enzyme fidelity) and nanoreactors (e.g., restricted reaction space for enzymes) have been proposed as ideal vehicles[16,17], owing to efficient codelivery of the non-toxic biological enzymes (e.g., GOx) and 1-MT to tumors. Thus, the in situ cargo release and substrate catalysis (e.g., glucose) lead to the generation of toxic species (e.g., $H_2O_2$) with improved bioavailability and therapy efficacy. On the other hand, the tumor microenvironment-activated size/charge changeable strategy can overcome these biological barriers, leading to the improved delivery efficiency and therapeutic effect[18,19]. Therefore, the MOF-based nanoreactors with the size/charge changeable features represent an appropriate nanosystem for boosting the antitumor immune response through starvation/oxidation combined IDO-blockade immunotherapy.

In this work, a pH/ROS dual-sensitive degradable MOF nanoreactor-based nanosystem (denoted as PCP-Mn-DTA@GOx@1-MT) with self-amplified drug release and enhanced tumor penetration has been constructed to co-deliver GOx and 1-MT for the tumor starvation/oxidation/IDO-blockade immunotherapy. Compared to previous studies, the present nanosystem exhibits following four important advantages (Fig. 1): (1) The tumor-activated degradable MOF nanoreactor is synthesized through the covalent crosslinking with the ROS-susceptible agents and $Mn^{2+}$, which can be rapidly disassembled triggered by the rich intracellular ROS of tumor cells, thus, minimizing the long-term retention toxicity of the conventional MOFs; (2) The size/charge changeable strategy designed in the nanoreactor sequentially breaks the biobarriers and improves the delivery efficiency. The shielding shell of the PCP-Mn-DTA@GOx@1-MT nanosystem exhibits a rapid removal of polyethylene glycol (PEG) component to afford a polyethylenimine (PEI)-conjugated cationic core in response to the weakly acidic tumor microenvironment (pH~6.8). The transformed nanosystem with strongly positive charge and small size significantly improves the tumor penetration depth and endocytosis; (3) The consumption of glucose by GOx is accompanied by the elevated generation of $H_2O_2$, which can be further converted to hydroxyl radical (·OH) with high toxicity through $Mn^{2+}$-mediated Fenton-like reaction, thus leading to the complete MOF degradation, drug release and improved therapeutic efficacy; (4) Taking advantage of the promoted immune response by GOx-mediated starvation/oxidation therapy and immune resistance suppression executed by IDO blockade immunotherapy, the PCP-Mn-DTA@GOx@1-MT nanosystem presents a remarkable therapeutic effect. Therefore, the successful construction of the multifunctional PCP-Mn-DTA@GOx@1-MT nanoreactor provides a paradigm for effectively overcoming the delivery biobarriers and revealing a superior tumor killing efficacy through starvation therapy along with immune modulatory effects.

## Results

### Synthesis and characterization of PCP-Mn-DTA@GOx@1-MT.
In order to construct ROS-responsive Mn-based MOF (Mn-DTA), a ROS-sensitive compound **1** (linker) was synthesized (see Methods section and Supplementary experimental details), which was confirmed by [1]H NMR and mass spectrometry (Fig. 2a, b). Mn-DTA was subsequently synthesized via the hydrothermal reaction. The strong diffraction peaks for 100°, 002°, 101°, 102°, 110°, 103° and 112° planes are observed in the powder X-ray diffraction (XRD) pattern (Fig. 2c), consistent with the crystal structure of MnS[20,21]. Furthermore, the calculated results based on the CCDC database demonstrated that $Mn^{2+}$ actually coordinates with three compound **1** ligands to form a regular hexagonal prism structure (Fig. 2d–f). In particular, three $Mn^{2+}$ ions occupy the three vertices of the hexagon, and two $Mn^{2+}$ ions locate at the two vertices of the quadrilateral in each hexagonal prism, whereas the linker between $Mn^{2+}$ and –COOH from compound **1** forms the sides of the hexagonal prism. Meanwhile, each $Mn^{2+}$ is shared by two adjacent quadrilaterals and one hexagon (Fig. 2g). These results collectively revealed that Mn-DTA possessed a three-dimensional porous network structure[20].

The morphological structure of Mn-DTA was detected by transmission/scanning electron microscopy (TEM/SEM). As shown in Fig. 3a and Supplementary Fig. 1a, Mn-DTA exhibits a spherical structure with a size of ~50 nm, which is beneficial for overcoming biobarriers of high interstitial pressure and dense tumor extracellular matrix, leading to a superior tumor penetration and an improved delivery efficiency[22,23]. The structure of Mn-DTA was further verified by energy-dispersive X-ray spectroscopy (EDS) elemental mappings of Mn, S and O (Fig. 3b), originating from $MnCl_2$ and compound **1**. Notably, Mn-DTA exhibits the lattice spacings of 0.305 and 0.328 nm detected by HRTEM (Supplementary Fig. 1b), which are consistent with the values for the (101, 002) planes of MnS determined by XRD. The Brunauer-Emmett-Teller (BET) analysis of Mn-DTA revealed a typical type IV isotherm (Supplementary Fig. 2), implying a distinct mesoporous structure. In addition, the surface area, average adsorption pore size and zeta potential of Mn-DTA were 107.8 $m^2 g^{-1}$, 8.61 nm and −54 mV (Supplementary Fig. 3 and Supplementary Table 1), respectively. The results confirmed that the Mn-DTA core with a suitable size was successfully synthesized.

Subsequently, PCP-Mn-DTA was constructed by conjugating compound **3** as the shell on the surface of the Mn-DTA core. Compound **3** was successfully synthesized as reported previously[19] and confirmed by [1]H NMR spectroscopy,

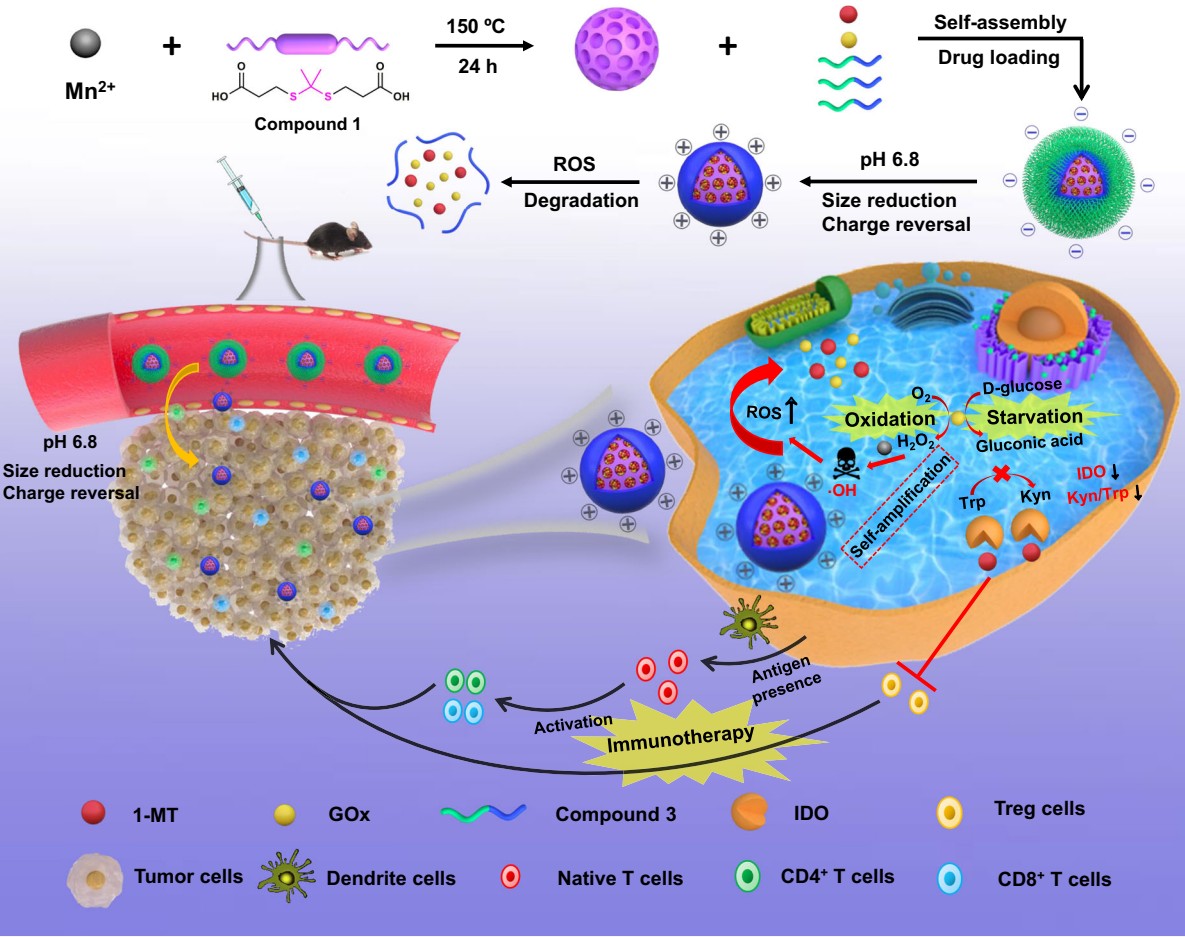

**Fig. 1 Construction of multifunctional nanoreactor and its immune stimulation capabilities.** Synthetic route and schematic illustration of PCP-Mn-DTA@GOx@1-MT nanosystem for combined starvation, oxidation and immunotherapy.

gel-permeation chromatography (GPC), and Fourier-transform infrared (FTIR) spectroscopy (Supplementary Figs. 4, 5 and Supplementary Table 2). After functionalization with the shell and co-loading with GOx and 1-MT, the obtained PCP-Mn-DTA@GOx@1-MT displayed a similar morphology as Mn-DTA core. However, an obvious outer layer existed around Mn-DTA, and the overall structure of PCP-Mn-DTA@GOx@1-MT became blurry (Fig. 3c). Furthermore, the diameter increased from $49.5 \pm 7.3$ to $97.0 \pm 4.5$ nm ($n = 300$), as confirmed by dynamic light scattering (DLS, Fig. 3d). Furthermore, the surface potential increased from $-54$ mV to $-13.7$ mV (Supplementary Fig. 3). Notably, the absorption spectrum of PCP-Mn-DTA@GOx@1-MT displayed the characteristic peaks of GOx and 1-MT (Fig. 3e), suggesting the successful drug loading. The loading contents of GOx and 1-MT were 8.8% and 13.5%, respectively (Supplementary Fig. 6). In addition, the nanosystem exhibited an effective biostability, as indicated by the size retention upon incubation with 10% serum for 6 days (Supplementary Fig. 7), which is conducive for drug delivery in vivo.

The vital features of the PCP-Mn-DTA nanosystem were the pH-sensitive size reduction, charge conversion, ROS-activated self-amplified degradation and cargo release, which are desired to enhance the tumor penetration and cellular uptake[24], acquire complete drug release, supply ROS for tumor killing, etc. The pH-responsive charge reversal and size reduction of the nanosystem were characterized by TEM, DLS and zeta potential. After the incubation with pH 6.8 (simulating tumor microenvironment), the average size of PCP-Mn-DTA@GOx@1-MT dramatically

decreased to around 55 nm (Fig. 3d), which was close to the size of native Mn-DTA, owing to the removal of the PEG shell[19]. Meanwhile, the surface potential accordingly reversed from $-13.7$ mV to $+34.2$ mV (Supplementary Fig. 3), thus confirming the pH-sensitive charge reversal.

The ROS-responsive self-amplified disassembly and drug release behavior of the nanosystem were verified by TEM, DLS and UV-vis spectroscopy. As demonstrated in Fig. 3f, g, the natural spherical structure of Mn-DTA and PCP-Mn-DTA@-GOx@1-MT collapsed upon exposure to $H_2O_2$, which was consistent with DLS (Fig. 3d), thus implying the ROS-responsive disassembly. In addition, the UV-vis spectra of PCP-Mn-DTA@GOx@1-MT revealed that the characteristic peak intensity of GOx and 1-MT enhanced upon increasing the concentration of $H_2O_2$ (Fig. 3e), indicating the ROS-responsive drug release. Notably, after exposed to glucose, the absorption peak intensity of GOx and 1-MT in the nanosystem was sharply enhanced, owing to the GOx catalyzed ROS self-generation and self-amplified drug release.

To further prove the ROS-sensitive drug release behavior, the real-time release assay was simultaneously conducted. A negligible GOx release (below 16%) was observed in the control group under physiological conditions (pH 7.4) for 24 h (Fig. 4a), suggesting the good stability of the nanosystem. In contrast, about 58% and 79% GOx was released from PCP-Mn-DTA@GOx@1-MT upon incubation with 0.1 and 1 mM $H_2O_2$ respectively, thus confirming the ROS-responsive drug release. After interaction with glucose, PCP-Mn-DTA@GOx@1-MT displayed a high GOx

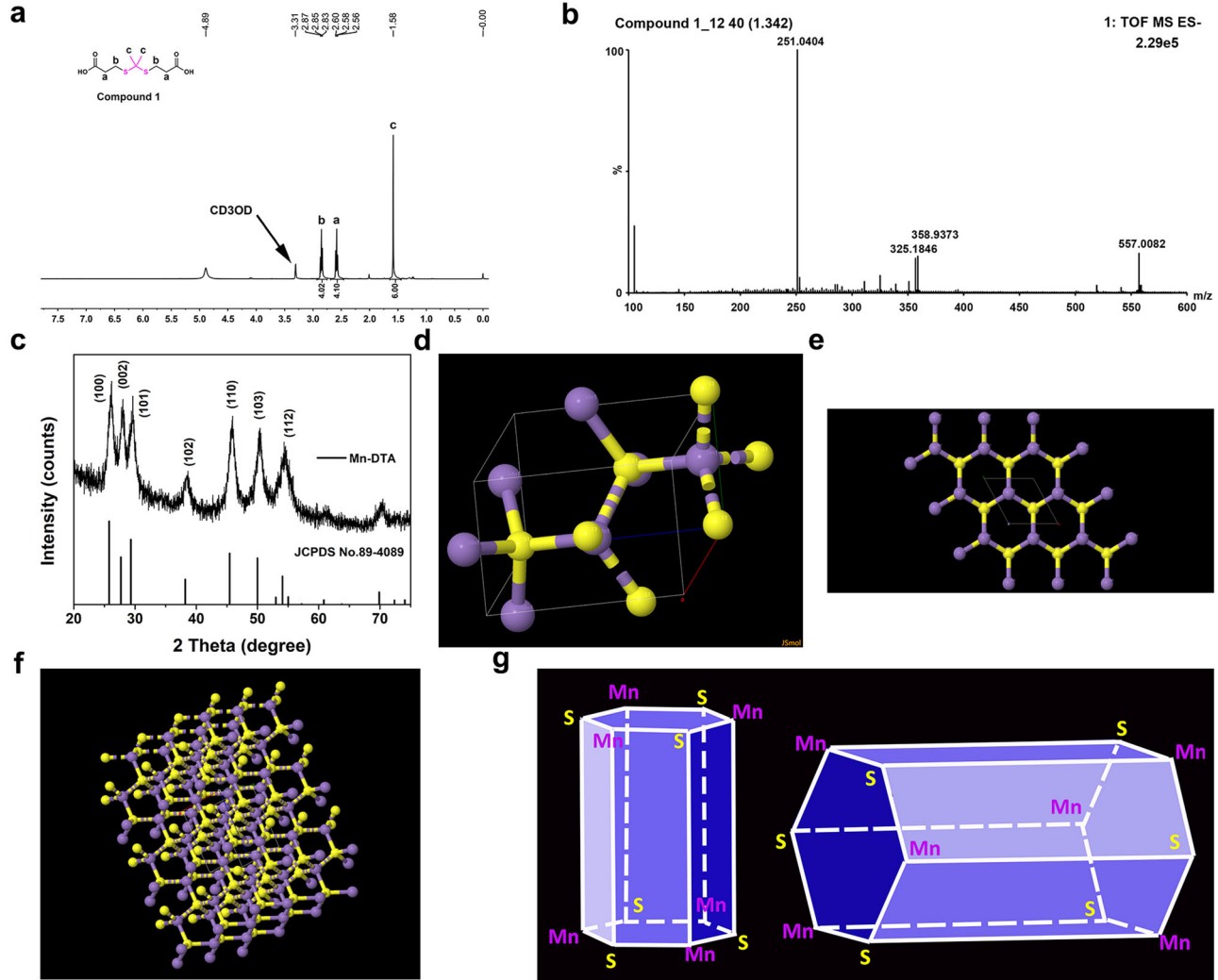

**Fig. 2 Characterization of Mn-DTA MOF. a** [1]H NMR and (**b**) ESI-MS spectra of compound **1**. **c** Powder XRD pattern of Mn-DTA (MnS, JCPDS No, 89-4089). **d** Unit cell structure and (**e**, **f**) crystalline structure (3 × 3 x 3) of MnS calculated by CCDC database (ICSD Entry: 44765). **g** Schematic diagram for the crystalline morphology of Mn-DTA-MOF. The purple and yellow joints represent Mn and S atoms, respectively.

release (~81% and 90%), due to the self-generation of ROS and cascade-amplified drug release triggered by GOx. It was worth noting that PCP-Mn-DTA@GOx@1-MT exhibited a sufficient drug release when exposed to the biologically relevant glucose and intratumoral ROS concentration (0.1 mM $H_2O_2$, Fig. 4a)[25]. This characteristic can potentially overcome the incomplete drug release limitation caused by the insufficient endogenous ROS concentration, thus improving tumor killing efficiency.

To reveal the catalytic activity of GOx and ROS self-generation capacity of the nanosystem, the enzyme activity test and ROS probe-based confocal laser scanning microscopy (CLSM) were employed[26]. Typically, free GOx and PCP-Mn-DTA@GOx@1-MT were incubated with glucose to evaluate the GOx activity, which was reflected by the pH change (induced by gluconic acid) and $H_2O_2$ generation. As shown in Fig. 4b, c, the pH values gradually decreased and $H_2O_2$ generation increased, along with the catalytic reaction duration, in both GOx and PCP-Mn-DTA@GOx@1-MT groups. The reduction of pH and generation of $H_2O_2$ in PCP-Mn-DTA@GOx@1-MT rapidly reached equilibrium within 1 h, and a similar trend was also observed in free GOx. These results confirmed a high catalytic activity of the PCP-Mn-DTA@GOx@1-MT nanosystem, which was not affected by the MOF encapsulation. Compared with the control and

PCP-Mn-DTA negative groups, both PCP-Mn-DTA@GOx and PCP-Mn-DTA@GOx@1-MT groups effectively generated ROS, as revealed by the green fluorescence and related quantitative analysis (Fig. 4d, e). The amount of ROS generation by PCP-Mn-DTA@GOx@1-MT drastically decreased in the absence of glucose ($p < 0.01$). The results confirmed that PCP-Mn-DTA@GOx@1-MT nanosystem could achieve an efficient drug release, and ROS generated by the nanosystem was indeed glucose-dependent.

**Cytotoxicity, tumor penetration and cellular uptake of PCP-Mn-DTA@GOx@1-MT.** The CCK8 assay was employed to evaluate the in vitro cytotoxicity of the PCP-Mn-DTA@GOx@1-MT nanosystem against B16F10 cells. Compared with the control, PCP-Mn-DTA exhibited a negligible cytotoxicity with dosages (50–1000 µg mL$^{-1}$) regardless of the incubation time (Supplementary Fig. 8), suggesting an optimal compatibility of the PCP-Mn-DTA nanocarrier. In addition, free 1-MT caused specific cell damage (Fig. 5a), owing to its weak toxicity[27]. Meanwhile, a relatively low cell viability was displayed for PCP-Mn-DTA@1-MT as compared to free 1-MT, inducing by the excellent drug delivery efficiency. After loading with GOx, PCP-Mn-DTA@-GOx@1-MT led to a more severe cell damage, attributed to the

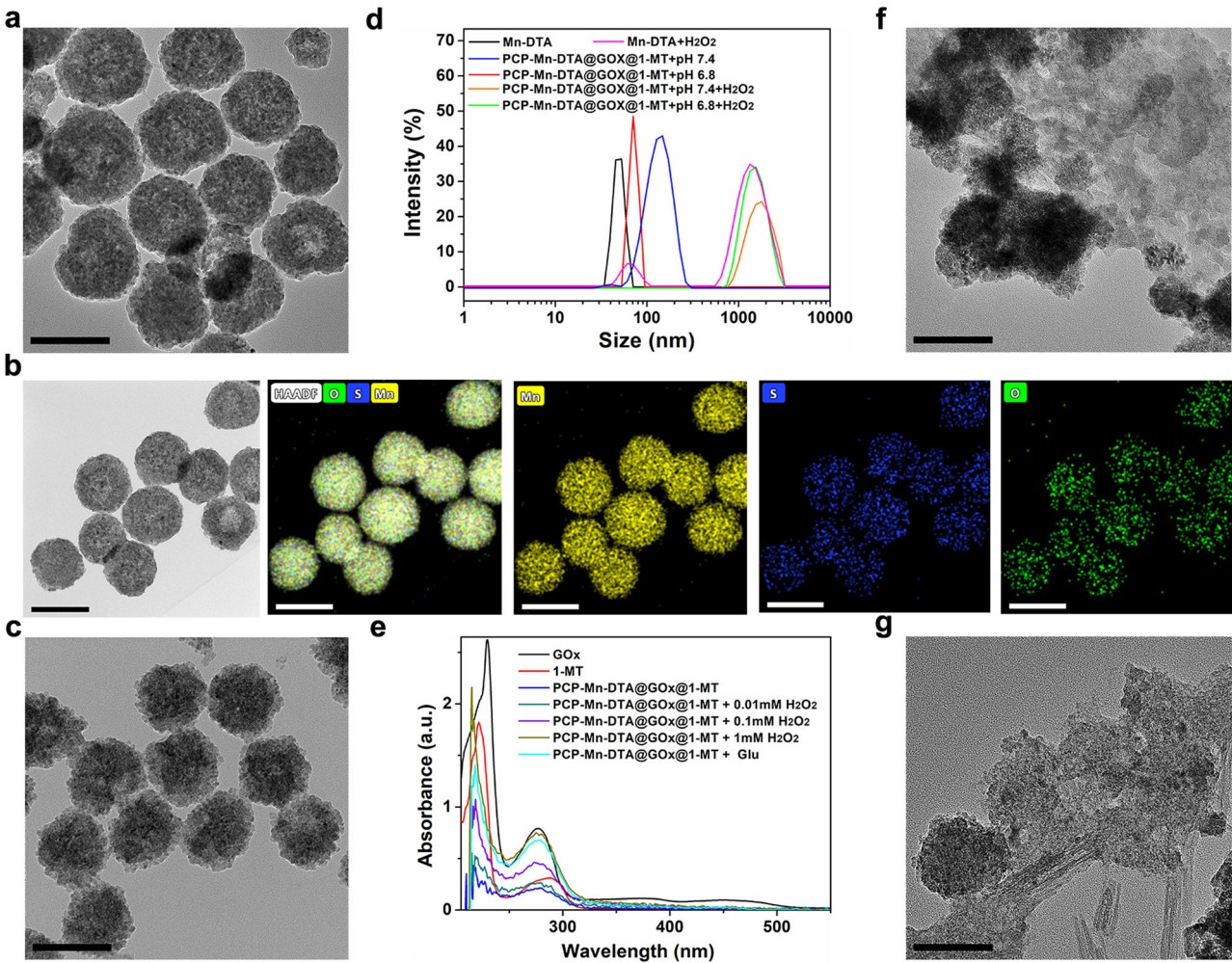

**Fig. 3 Characterization of PCP-Mn-DTA@GOx@1-MT nanosystem. a** TEM image and (**b**) EDS element mappings of Mn-DTA. **c** TEM image of PCP-Mn-DTA@GOx@1-MT. **d** DLS curves of Mn-DTA and PCP-Mn-DTA@GOx@1-MT after respective treatment at pH 7.4 or 6.8 with or without $H_2O_2$ for 4 h. **e** Absorbance spectra of GOx, 1-MT and PCP-Mn-DTA@GOx@1-MT incubated with various concentrations of $H_2O_2$ or glucose for 12 h. **f, g** TEM images of Mn-DTA (**f**) and PCP-Mn-DTA@GOx@1-MT (**g**) treated with $H_2O_2$ for 4 h. Scale bars: 50 nm for **a**, **b** and **f**, 100 nm for **c** and **g**.

GOx-induced starvation/oxidation damage[28,29]. Moreover, the lowest cell viability occurred in PCP-Mn-DTA@GOx@1-MT plus glucose after 48 h post-incubation. The results indicated that the nanosystem effectively killed the tumor cells through GOx-catalyzed ROS generation (Fig. 4d, e), and the addition of glucose further promoted the B16F10 cell death via GOx catalysis.

The B16F10 cell-based multicellular spheroids (MCSs) were constructed for evaluating the tumor infiltration property of the nanosystem[19]. For this purpose, GOx and PCP-Mn-DTA@-GOx@1-MT were labelled with fluorescein isothiocyanate (FITC), and CLSM was used to monitor the penetration of the nanosystem. As shown in Fig. 5b, PCP-Mn-DTA@GOx@1-MT was dispersed around the outer boundary of MCSs at pH 7.4 after 4 h post-incubation, and only a small fraction of PCP-Mn-DTA@GOx@1-MT entered inside MCSs upon 12 h incubation, indicating the restricted penetration of PCP-Mn-DTA@GOx@1-MT owing to the relatively large size. Interestingly, the green fluorescence emitted by PCP-Mn-DTA@GOx@1-MT was widely distributed over the whole MCSs after the incubation at pH 6.8 for 4 h, suggesting the good tumor penetration. Moreover, the penetration tendency increased further as the incubation duration was prolonged to 12 h, as revealed by much stronger green fluorescence. The quantitative analysis confirmed the

pH-enhanced tumor penetration of the nanosystem (Supplementary Fig. 9), due to the size reduction and charge conversion in response to the weakly acidic tumor microenvironment. The results collectively demonstrated that PCP-Mn-DTA@GOx@1-MT nanosystem possessed favorable tumor penetration capability, which is helpful for overcoming biobarriers and improving the drug delivery efficiency.

The cellular uptake level of the nanosystem in the B16F10 cells was subsequently investigated by flow cytometry (FCM) and CLSM. As compared to the control, the uptake of PCP-Mn-DTA@GOx@1-MT by the B16F10 cells exhibited a time-dependent behavior, and the endocytosis amount of PCP-Mn-DTA@GOx@1-MT was remarkably higher than free GOx at pH 7.4 or pH 6.8 (Fig. 5c, d), indicating again the superior delivery effect of PCP-Mn-DTA nanocarrier. Furthermore, the endocytosed population at pH 6.8 was much higher than that at pH 7.4 regardless of the incubation time. This phenomenon is due to the fact that the size and charge transformation of the nanosystem at a low pH increases the endocytosis efficiency. These results indicated that the size reduction and charge conversion of the nanosystem activated by the weakly acidic tumor microenvironment could significantly enhance the tumor infiltration and cellular uptake, which is helpful for killing tumors[30,31].

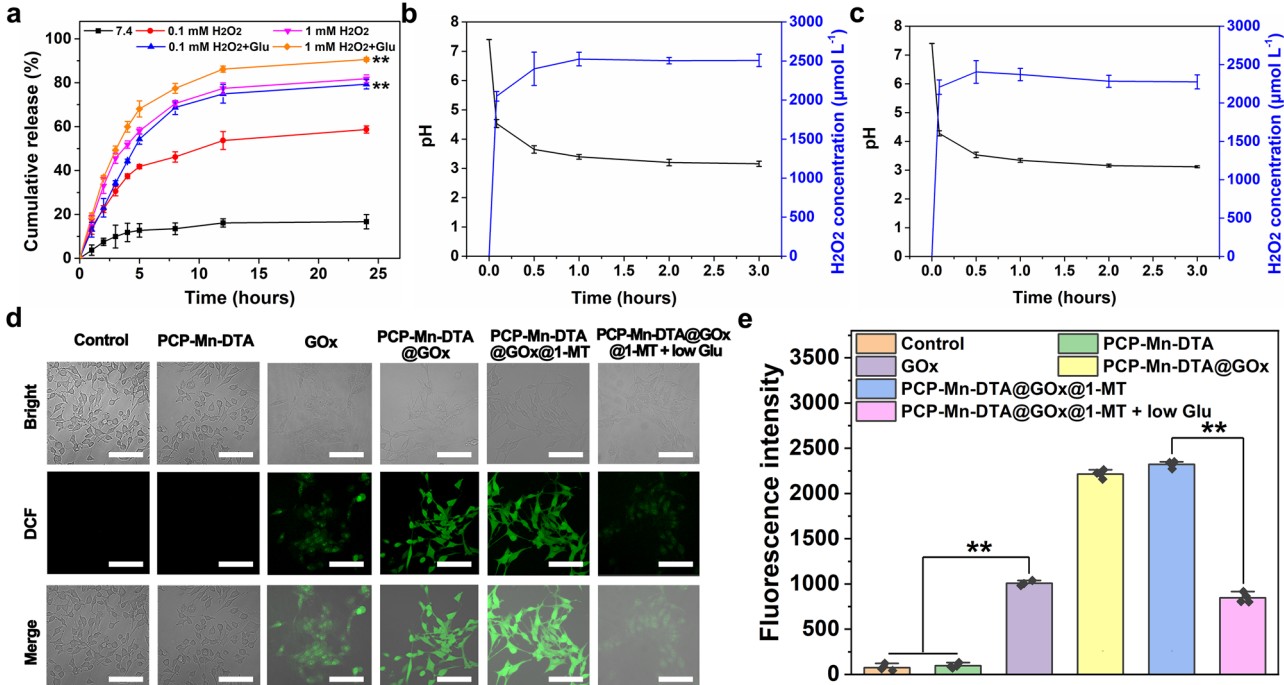

**Fig. 4 Functional properties of PCP-Mn-DTA@GOx@1-MT nanosystem. a** Cumulative release of GOx from PCP-Mn-DTA@GOx@1-MT after treatments with different concentrations of $H_2O_2$ or $H_2O_2$ plus glucose (abbreviated as Glu). **b**, **c** pH values and generated $H_2O_2$ concentrations at various time points arisen from GOx (**b**) and PCP-Mn-DTA@GOx@1-MT (**c**) catalyzed disintegration reaction of glucose. **d** CLSM images and (**e**) quantitative measurements of intracellular ROS. Data represent mean ± SD ($n = 4$ independent samples). P-values were determined by unpaired Student's t-test (two-tailed), **$p < 0.01$. Scale bar: 50 μm.

**IDO enzyme inhibition effect of PCP-Mn-DTA@GOx@1-MT.** To study the inhibitory effect of the nanosystem on IDO enzyme activity in vitro, the expression of IDO in the B16F10 cells treated with PCP-Mn-DTA@GOx@1-MT was measured using western blotting. Compared with the control (PBS) and PCP-Mn-DTA negative groups, 1-MT displayed an obvious IDO expression inhibition (Fig. 6a)[32]. Furthermore, both PCP-Mn-DTA@1-MT and PCP-Mn-DTA@GOx@1-MT inhibited the downregulation of IDO expression more significantly as compared to free 1-MT ($p < 0.01$, Fig. 6b), suggesting that the IDO activity suppression was benefited by the advanced delivery efficiency of the nanosystem[10].

In order to reveal the IDO inhibition effect mediated by the PCP-Mn-DTA@GOx@1-MT nanosystem on the T cell proliferation, an in vitro co-culture model composed of the B16F10 cells and lymphocytes was constructed to examine the proliferation[33]. As shown in Fig. 6c, compared to the control (PBS), the proportion of the EdU+ T cells was significantly increased after treatment with 1-MT and 1-MT loaded formulations (PCP-Mn-DTA@1-MT and PCP-Mn-DTA@GOx@1-MT), thus, implying the improved T cell proliferation. Moreover, the amount of the proliferated T cells in PCP-Mn-DTA@GOx@1-MT was larger than the 1-MT group (Fig. 6d), thus reconfirming the excellent delivery efficiency of the nanosystem. The phenomena can be explained as follows (Fig. 6e): first, the uptake efficiency and bioavailability of free 1-MT are limited by its poor solubility. In contrast, PCP-Mn-DTA with a high drug loading capacity can effectively overcome the poor solubility of 1-MT and improve the drug bioavailability. Second, PCP-Mn-DTA@GOx@1-MT can be effectively endocytosed by the B16F10 cells to release 1-MT and GOx in response to the intracellular ROS and additional ROS generated by GOx. Thus, an abundant of 1-MT could significantly inhibit IDO and recover the T cell activity. These results confirm the significant potential of the nanosystem to enhance the antitumor immunity in vivo.

**Lysosomal escape, cell apoptosis and anti-metastasis of PCP-Mn-DTA@GOx@1-MT in vitro.** The lysosomal escape capability of the PCP-Mn-DTA@GOx@1-MT nanosystem is vital for maintaining the bioactivity of GOx and achieving the desired tumor damage. CLSM was used to monitor the lysosomal escape of the nanosystem, determined from the degree of match between the green fluorescence (FITC labelled nanosystem) and red fluorescence (lysotracker red marked lysosomes)[34]. On incubating the B16F10 cells with PCP-Mn-DTA@GOx@1-MT for 1 h, the green fluorescence marked PCP-Mn-DTA@GOx@1-MT was noted to be largely distributed at the edge of the cell membrane (Fig. 7a), demonstrating a natural sign for endocytosis. As the incubation duration was prolonged to 3 h, almost all PCP-Mn-DTA@-GOx@1-MT was localized in the lysosomes, as revealed by the well-matched bright yellow fluorescence. Furthermore, the green fluorescence marked PCP-Mn-DTA@GOx@1-MT was transferred from the lysosomes to the cytoplasm as the incubation duration was prolonged to 8 h, indicating a successful lysosome escape. The PEI-functionalized nanosystem could induce the lysosomal membrane disruption through the inherent protonation of PEI[35,36], thereby resulting in the lysosome escape as well as the cytoplasmic release of GOx and 1-MT for killing the tumor cells.

Subsequently, FCM and CLSM were employed to accurately monitor the apoptosis/death ratio of the B16F10 cells incubated with the various samples using Annexin V-FITC/PI (PI: propidium iodide) and live-dead staining kits. As shown in Fig. 7b, compared with the control, a low degree of apoptosis and death ratio (6.58%) were observed from the blank vehicle PCP-Mn-DTA upon incubation for 24 h, implying its superior compatibility. Free 1-MT caused a moderate degree of apoptosis, which was obviously lower than that of PCP-Mn-DTA@1-MT (12.91% vs. 21.7%, $p < 0.01$, Fig. 7c), thanks to the natural cytotoxicity of 1-MT[37] and the delivery effect of the nanosystem.

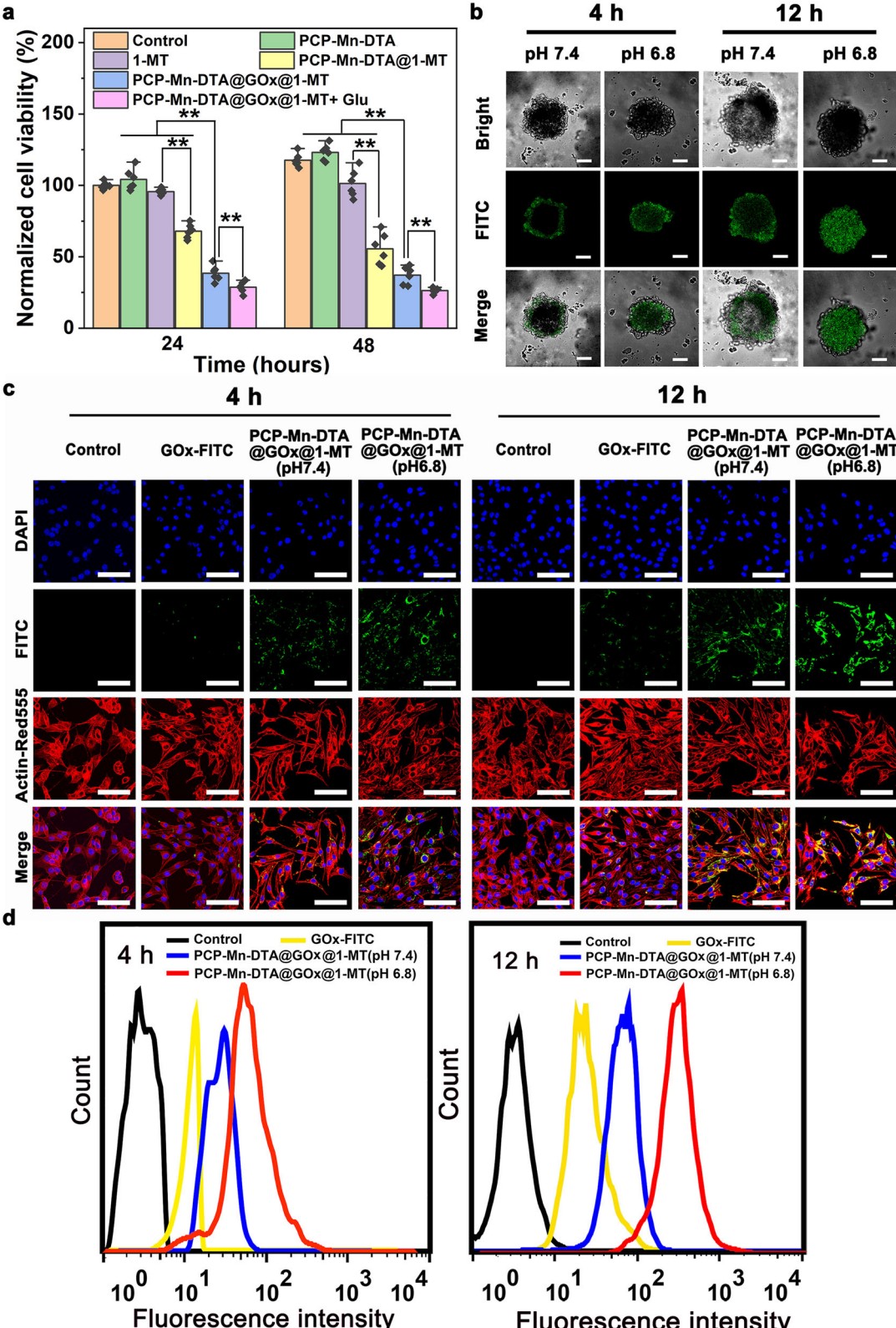

**Fig. 5 In vitro cytotoxicity, cellular uptake, and tumor penetration studies of PCP-Mn-DTA@GOx@1-MT nanosystem. a** Cytotoxicity studies of B16F10 cells cultivating with PCP-Mn-DTA, 1-MT, PCP-Mn-DTA@1-MT, and PCP-Mn-DTA@GOx@1-MT with or without glucose for 24 h and 48 h, respectively. **b** Tumor penetration and (**c**) endocytosis images of PCP-Mn-DTA@GOx@1-MT labelled with FITC in B16F10 MCSs after respective incubation at pH 6.8 or 7.4 for 4 h and 12 h, with images representative of 3 experiments. Nuclei and cytoskeleton were respectively stained with DAPI (blue) and ActinRed™555 (red). **d** Quantitative FCM analysis based on (**c**). Data represent mean ± SD ($n = 6$ biologically independent samples). $P$-values were determined by unpaired Student's $t$-test (two-tailed), \*\*$p < 0.01$. Scale bars: 100 µm for **b**, 50 µm for **c**.

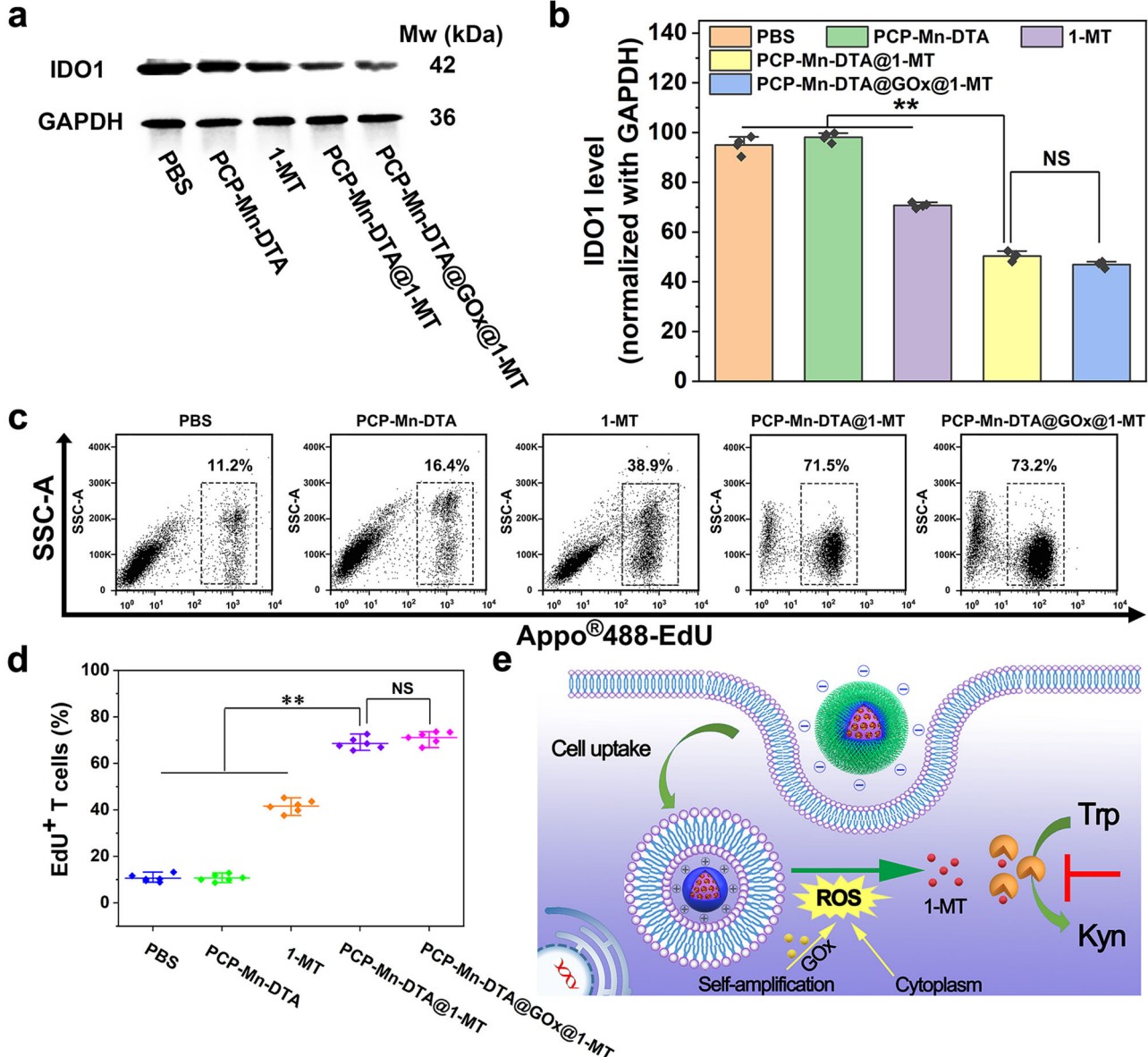

**Fig. 6 IDO inhibition effect of PCP-Mn-DTA@GOx@1-MT nanosystem. a, b** Effect of PCP-Mn-DTA@GOx@1-MT on the IDO expression after the treatment as detected by western blot (**a**) and quantitative analysis (**b**). **c** FCM results and (**d**) quantitative analysis on the EdU$^+$ T cell proportion cocultured with B16F10 cells after the treatments with PBS, PCP-Mn-DTA, 1-MT, PCP-Mn-DTA@1-MT and PCP-Mn-DTA@GOx@1-MT, respectively. **e** Schematic representation of PCP-Mn-DTA@GOx@1-MT nanosystem on the inhibition mechanism of IDO1 in tumor cells. Data shown as mean ± SD and representative of 4 (**a, b**) or 6 (**c, d**) independent experiments. *P*-values were determined by unpaired Student's *t*-test (two-tailed), **$p < 0.01$.

PCP-Mn-DTA@GOx@1-MT led to a severe apoptosis (47.4%), resulting from the GOx-generated starvation/oxidation damage. Obvious $H_2O_2$ generation and inhibition of the ATP production by the GOx-loaded systems (PCP-Mn-DTA@GOx and PCP-Mn-DTA@GOx@1-MT, Supplementary Fig. 10) were observed. Moreover, PCP-Mn-DTA@GOx@1-MT plus glucose generated the highest apoptosis ratio in all treatments (54.1%), and the live-dead assay also reflected the similar tumor killing tendency (Supplementary Fig. 11). Above results comprehensively confirmed the superior tumor killing efficiency of the nanosystem.

To further illustrate the apoptosis mechanism induced by the PCP-Mn-DTA@GOx@1-MT nanosystem, the western blot assay was employed to evaluate the expression of the related apoptosis proteins, including Bax, Bcl-2 and Cyt-C. As demonstrated in Fig. 7d, a faint upregulation of the Bax/Bcl-2 ratio and Cyt-C was observed in free 1-MT group as compared to the control, while

PCP-Mn-DTA@1-MT induced more obvious expression tendency than that of 1-MT, confirming the initial cytotoxicity of 1-MT and intensive delivery efficiency of the nanosystem. Furthermore, PCP-Mn-DTA@GOx@1-MT led to a conspicuous upregulation of Bax and Cyt-C as well as a downregulation of Bcl-2, attributed to the combined effect of the GOx-mediated starvation/oxidation damage and 1-MT-caused cytotoxicity. PCP-Mn-DTA@GOx@1-MT plus glucose induced the highest upregulation of Bax/Bcl-2 and Cyt-C expression among all groups (Fig. 7e, f), thus, indicating the superior antitumor effect caused by PCP-Mn-DTA@GOx@1-MT.

As the energy metabolism regulated by GOx and 1-MT was closely associated with the tumor migration[38,39], the degree of anti-migration of PCP-Mn-DTA@GOx@1-MT was evaluated by the wound healing, tumor migration and invasion studies. As shown in Fig. 7g, the control and PCP-Mn-DTA groups exhibited

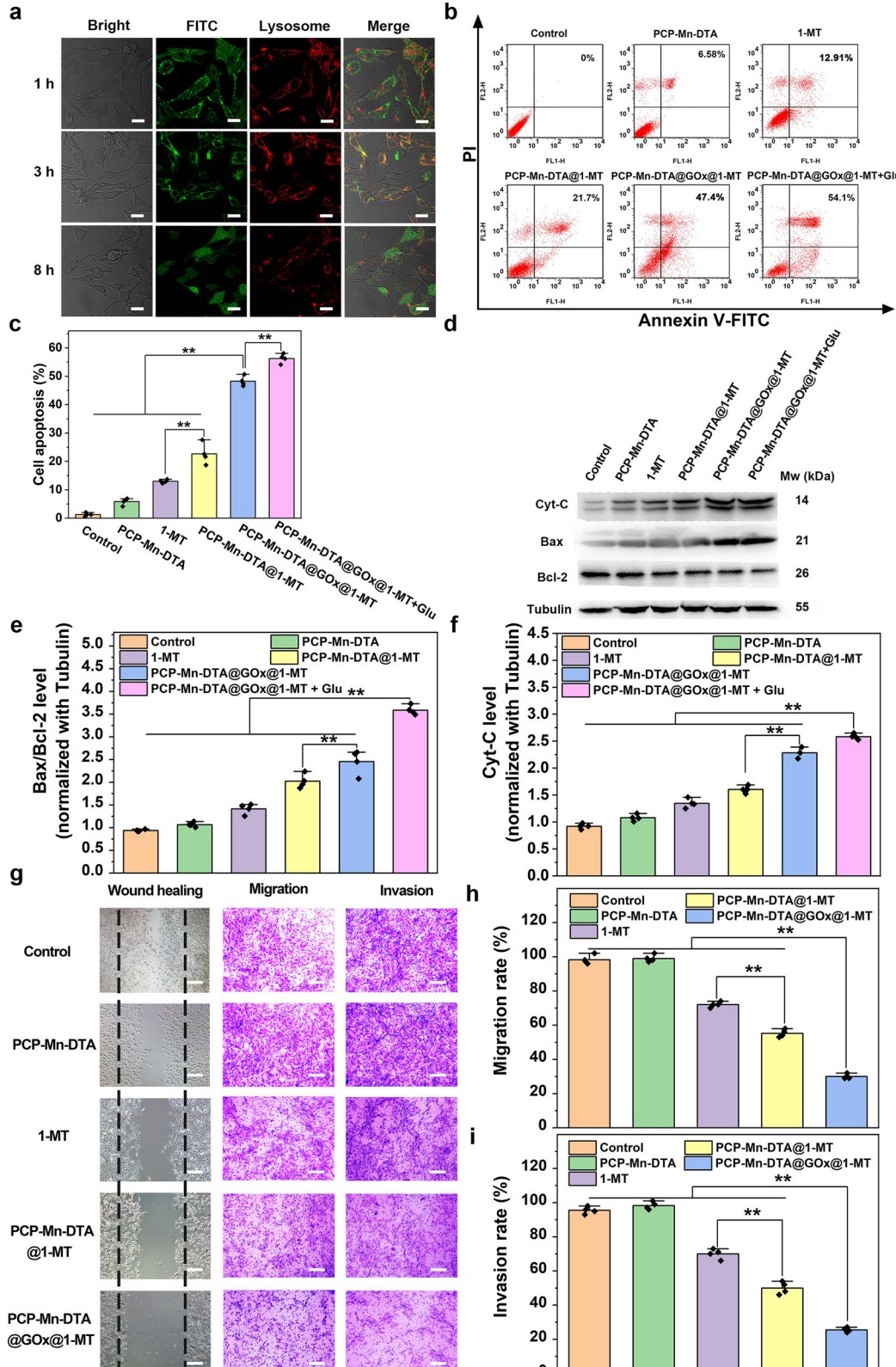

**Fig. 7 In vitro lysosomal escape, cell apoptosis and anti-metastasis studies of PCP-Mn-DTA@GOx@1-MT nanosystem. a** B16F10 cells incubated with FITC-labelled PCP-Mn-DTA@GOx@1-MT for different times, followed by staining with Lysotracker red to image the lysosome escape, with images representative of 3 experiments. **b**, **c** FCM and quantitative apoptosis analysis of B16F10 cells after difference treatments for 24 h. **d**–**f** Apoptosis-associated proteins in B16F10 cells after the treatment with indicated systems examined by western blot (**d**), and the related quantitative analysis of Bax/Bcl-2 (**e**) and Cytochrome C (**f**). **g**–**i** Microscopy images (**g**) and quantitative analysis of the wound healing and migration (**h**) as well as invasion assays (**i**) of B16F10 cells after the treatment with these systems. Data represent mean ± SD ($n = 4$ biologically independent samples). P-values were determined by unpaired Student's t-test (two-tailed), **$p < 0.01$. Scale bars: 20 μm for **a**, 100 μm for **g**.

a strong healing behavior, reflected by the disappearance of the scratches. It suggested that the B16F10 cells had inherent metastatic characteristics. 1-MT and PCP-Mn-DTA@1-MT demonstrated a modest anti-migration effect with healing rates of 75% and 58%, respectively (Supplementary Fig. 12). The PCP-Mn-DTA@GOx@1-MT group induced the lowest healing rate of 39%, which indicated the most effective cell motility inhibition. Moreover, PCP-Mn-DTA@GOx@1-MT also displayed the strongest migration and invasion suppression with coverage rates of 30.1% and 24.9%, respectively ($p < 0.01$, Fig. 7h, i), as further revealed by the expression and quantitative analysis of the tumor invasion-associated proteins (E-Cadherin, MMP2, and MMP9, Supplementary Fig. 13). As a result, high antitumor and anti-metastasis effect in vitro were obtained owing to the combined action of the GOx-mediated starvation/oxidation damage, 1-MT-caused cytotoxicity and immune regulation, which were further supported by the subsequent in vivo immune response assays.

**In vivo immune response reinforcement and immune evasion inhibition of PCP-Mn-DTA@GOx@1-MT.** In order to validate the antitumor immune response of the PCP-Mn-DTA@GOx@1-MT nanosystem, the relevant immune T cells at the tumor sites were quantitatively analyzed by FCM in the B16F10 and 4T1 tumor-bearing mice. As demonstrated in Fig. 8a–d and Supplementary Figs. 14 and 15, a large amount of Treg cells and a small extent of tumor-infiltrating cytotoxic T cells (CTLs) were observed in the control group, attributed to the innate immune resistance. Meanwhile, the amount of CTLs exhibited a moderate increase, whereas the Treg cells displayed a small decrease, in free 1-MT group. PCP-Mn-DTA@1-MT demonstrated a superior tendency than free 1-MT, which was attributed to the relieved effector T cell activity suppression and inhibition of the Treg cell proliferation by IDO blocking[40,41] as well as high drug delivery efficiency. Remarkably, PCP-Mn-DTA@GOx@1-MT induced the highest extent of tumor-infiltrating CTLs and lowest amount of Treg cells regardless of the B16F10 and 4T1 tumor-bearing mouse models ($p < 0.01$, Fig. 8b, d), indicating the reinforced immune response activation with a suppressed immune evasion. Notably, the proportion of the CD4[+] helper T cells in the free 1-MT and 1-MT loaded groups decreased in the B16F10 tumor-bearing mice (Supplementary Fig. 16), which was consistent with the previously reported studies[9,42].

IDO1 can catalyze the metabolism of Trp to Kyn, thus, leading to the direct inhibition of the mammalian target of rapamycin (mTOR) and activation of the signal transducers/activators of transcription (STAT3) through the AHR-IL-6-STAT3 pathway[43–45]. The ratio of Kyn to Trp, expression of intratumoral IDO1, mTOR and STAT3 can act as indicators to reveal the relieved immune resistance mechanism of the PCP-Mn-DTA@GOx@1-MT nanosystem. As shown in Fig. 8e, f, both PCP-Mn-DTA@1-MT and PCP-Mn-DTA@GOx@1-MT effectively inhibited the IDO expression, as manifested by the lowest ratio of intratumor Kyn to Trp as well as a significant downregulation of IDO1. In addition, an obvious upregulation of mTOR and a downregulation of STAT3 were observed in PCP-Mn-DTA@1-MT and PCP-Mn-DTA@GOx@1-MT groups, thus, implying that the 1-MT loaded nanosystem effectively blocked the IDO1 mediated immune resistance and enhanced the immune response.

As the effective tumor cell killing is positively associated with the level of the matured dendritic cells as well as the reinforcement of the immune response activation by exposing TAAs and presenting the "eat me" signal[46], the proliferation of the matured dendritic cells was investigated to further illustrate the mechanism of the nanosystem for boosting the immune response. As demonstrated in Fig. 8b, d and Supplementary

Fig. 17, 1-MT, PCP-Mn-DTA@1-MT and PCP-Mn-DTA@-GOx@1-MT promoted the dendritic cell maturation, with the sequence 1-MT < PCP-Mn-DTA@1-MT < PCP-Mn-DTA@-GOx@1-MT, thus, confirming the effective recruitment of the dendritic cells. Excitingly, PCP-Mn-DTA@GOx@1-MT also presented the highest recruitment of the intratumoral NK and B cells in either B16F10 or 4T1 tumor-bearing mice ($p < 0.01$, Fig. 8b, d, and Supplementary Figs. 18 and 19). It could be suggested that PCP-Mn-DTA@GOx@1-MT could effectively induce the tumor damage via the GOx-mediated starvation/oxidation therapy (Fig. 5a and Fig. 7b–f). Thus, the enhanced tumor immunogenicity and exposure of TAAs significantly reinforced the intratumoral recruitment of the effector T cells, and the 1-MT-initiated immune resistance suppression (Fig. 6, and 8b, d) further boosted the overall antitumor immune response. By taking advantage of the GOx-triggered starvation/oxidation therapy and 1-MT-mediated IDO-blockade immunotherapy, the PCP-Mn-DTA@GOx@1-MT nanosystem effectively strengthened the systemic antitumor immune response.

**In vivo antitumor effect of PCP-Mn-DTA@GOx@1-MT.** To authenticate the advantage of the combined starvation/oxidation therapy and IDO-blockade immunotherapy, the antitumor evaluation was simultaneously performed in vivo in both B16F10 and 4T1 tumor-bearing mouse models. In these models, the PCP-Mn-DTA group exhibited a rapid tumor growth similar to the control (Fig. 9a, a'), indicating the negligible antitumor effect of the empty carrier. Furthermore, the free 1-MT and PCP-Mn-DTA@1-MT groups exhibited slight and moderate tumor growth inhibition, attributed to the individual immunotherapy and advanced delivery efficiency. More importantly, the PCP-Mn-DTA@GOx@1-MT group induced the most obvious tumor growth inhibition without a significant body weight loss (Supplementary Fig. 20), thus, indicating the most effective antitumor therapy with reduced side effects. The tumor volume analysis (Fig. 9b, b') further confirmed the similar trend of the tumor growth inhibition. Additionally, the PCP-Mn-DTA@GOx@1-MT group remarkably prolonged the survival time of the tumor-bearing mice beyond 30 days (67% and 50% in the B16F10 and 4T1 tumor models, Fig. 9c, c'), which was much higher than the other treatment groups. Notably, PCP-Mn-DTA@GOx@1-MT demonstrated the strongest inhibitory effect on the lung metastasis as well (Fig. 9d, d'), as verified by the corresponding hematoxylin and eosin (H&E) staining of the lung tissues and subsequent quantification analysis (Fig. 9e, e').

The excellent antitumor and anti-metastasis effects of the PCP-Mn-DTA@GOx@1-MT nanosystem could be explained as follows. Firstly, the nanosystem endowed with the pH-responsive size/charge transformation could markedly increase the tumor permeability and cellular uptake, thereby improving the delivery efficacy and bioavailability of GOx and 1-MT. Secondly, GOx from PCP-Mn-DTA@GOx@1-MT could effectively kill the tumor cells and expose TAAs via starvation/oxidation[47], leading to effective recruitment of the effector T cells. Furthermore, 1-MT from PCP-Mn-DTA@GOx@1-MT could significantly suppress the IDO expression and relieve IDO-mediated immune tolerance, resulting in the recruitment of CTLs, B cells and NK cells as well as the inhibition of Treg cell proliferation. Lastly, the PCP-Mn-DTA@GOx@1-MT nanosystem combined with the starvation/oxidation therapy and IDO-blockade immunotherapy provided the most significant antitumor effect with reinforced immune activation and weakened immune tolerance.

Subsequently, H&E, terminal deoxynucleotidyl transferase dUTP nick end labeling (TUNEL) and immunofluorescence

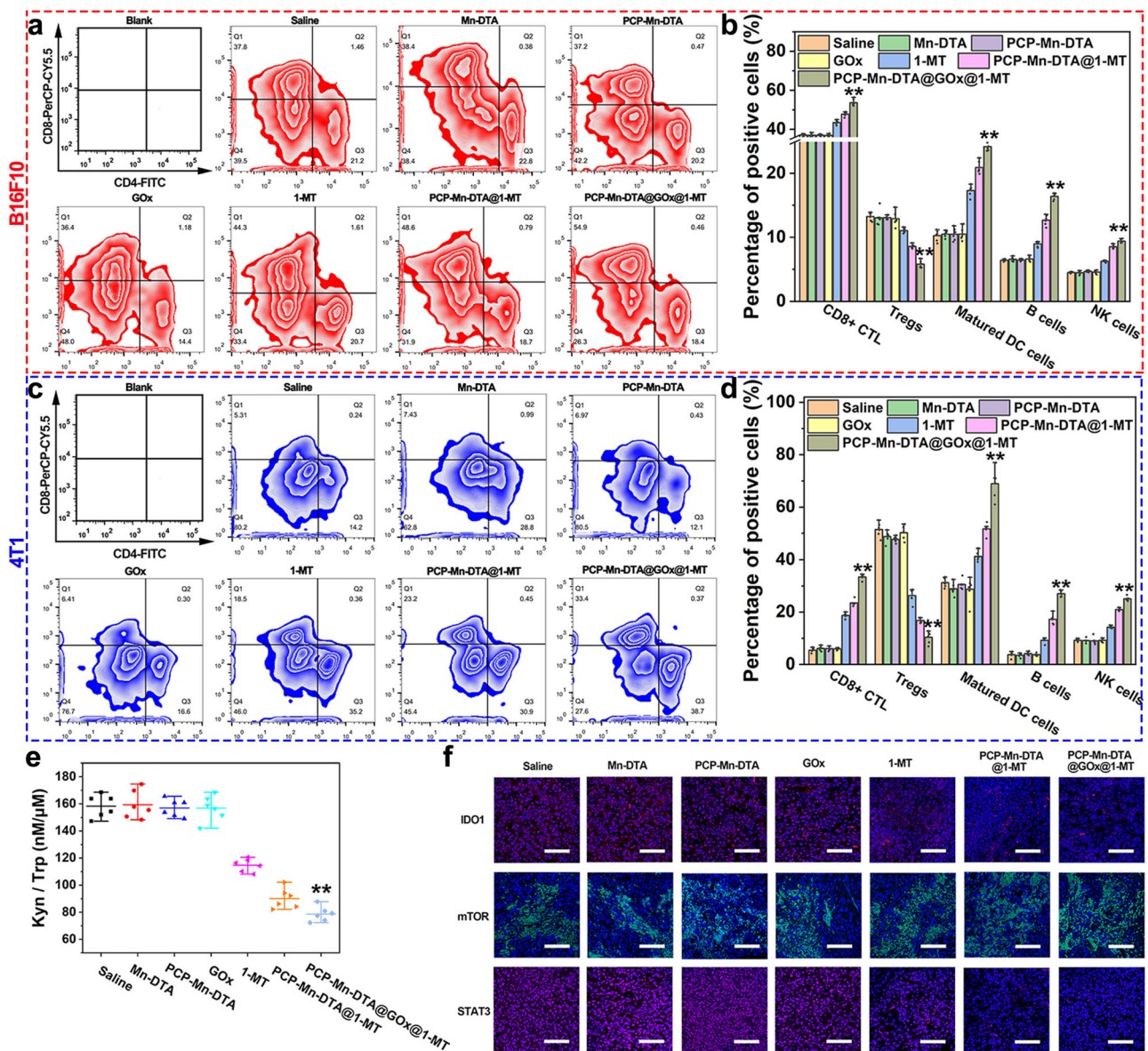

**Fig. 8 In vivo activation of immune response and suppression of immune resistance by PCP-Mn-DTA@GOx@1-MT nanosystem. a, c** FCM and
(**b, d**) quantitative analysis on the populations of tumor-infiltrating CD8$^+$ T cytotoxic cells, Treg cells, NK cells, dendritic cells, and B cells from the B16F10
tumor-bearing C57BL/6 mice and 4T1 tumor-bearing Balb/c mice after 4 days post-administration, respectively. **e** Kyn/Trp ratio changes of B16F10 tumor
model after above treatment for 4 days. **f** IFC images of tumor sections stained with anti-IDO, anti-mTOR, and anti-STAT3, with images representative of 3
experiments. Scale bar: 100 μm. Data represent mean ± SD ($n = 6$ biologically independent samples). $P$-values were determined by unpaired Student's $t$-
test (two-tailed), $^{**}p < 0.01$.

(IFC) staining analysis were conducted to further prove the
comprehensive antitumor activity of the nanosystem in vivo.
PCP-Mn-DTA@GOx@1-MT induced the most severe tumor
damage, as revealed by the distinct tissue dissociation in the H&E
images and numerous magenta dots in the TUNEL observation
(Fig. 9f, f'). In addition, the IFC analysis of Ki67 in the tumor
tissues confirmed that PCP-Mn-DTA@GOx@1-MT effectively
down-regulated the Ki67 expression, confirming its superior
antitumor effect.

The noninvasive photoacoustic bioimaging was performed for
measuring the intratumoral blood oxygen level (sO2 average) and
H$_2$O$_2$ production after intravenous injection of PCP-Mn-
DTA@GOx@1-MT. The photoacoustic images and quantitative
results indicated that the sO2 average displayed a ~81% drop after
the administration of PCP-Mn-DTA@GOx@1-MT for 24 h, and

the photoacoustic imaging intensity from the tumor site was
correspondingly enhanced (Fig. 9g and Supplementary Fig. 21).
In comparison, the GOx group exhibited no significant change.
The reason could be explained as follows: (1) the intratumoral
ROS accelerated the GOx release from the PCP-Mn-DTA@-
GOx@1-MT nanosystem, and the released GOx consumed the
endogenous oxygen and "starved" tumor, thus, resulting in the
aggravation of the local acidity and H$_2$O$_2$ level; (2) the GOx-
induced generation of H$_2$O$_2$ further amplified the disassembly of
the nanosystem and GOx release through a cascade reaction; (3)
the ·OH radical generated from the Fenton-like reaction between
Mn$^{2+}$ and H$_2$O$_2$ contributed to the augmented photoacoustic
signal as well[48]. Notably, the quantitative analysis of melanin in
mice demonstrated that PCP-Mn-DTA@GOx@1-MT signifi-
cantly reduced the amount of intratumoral melanin, while there

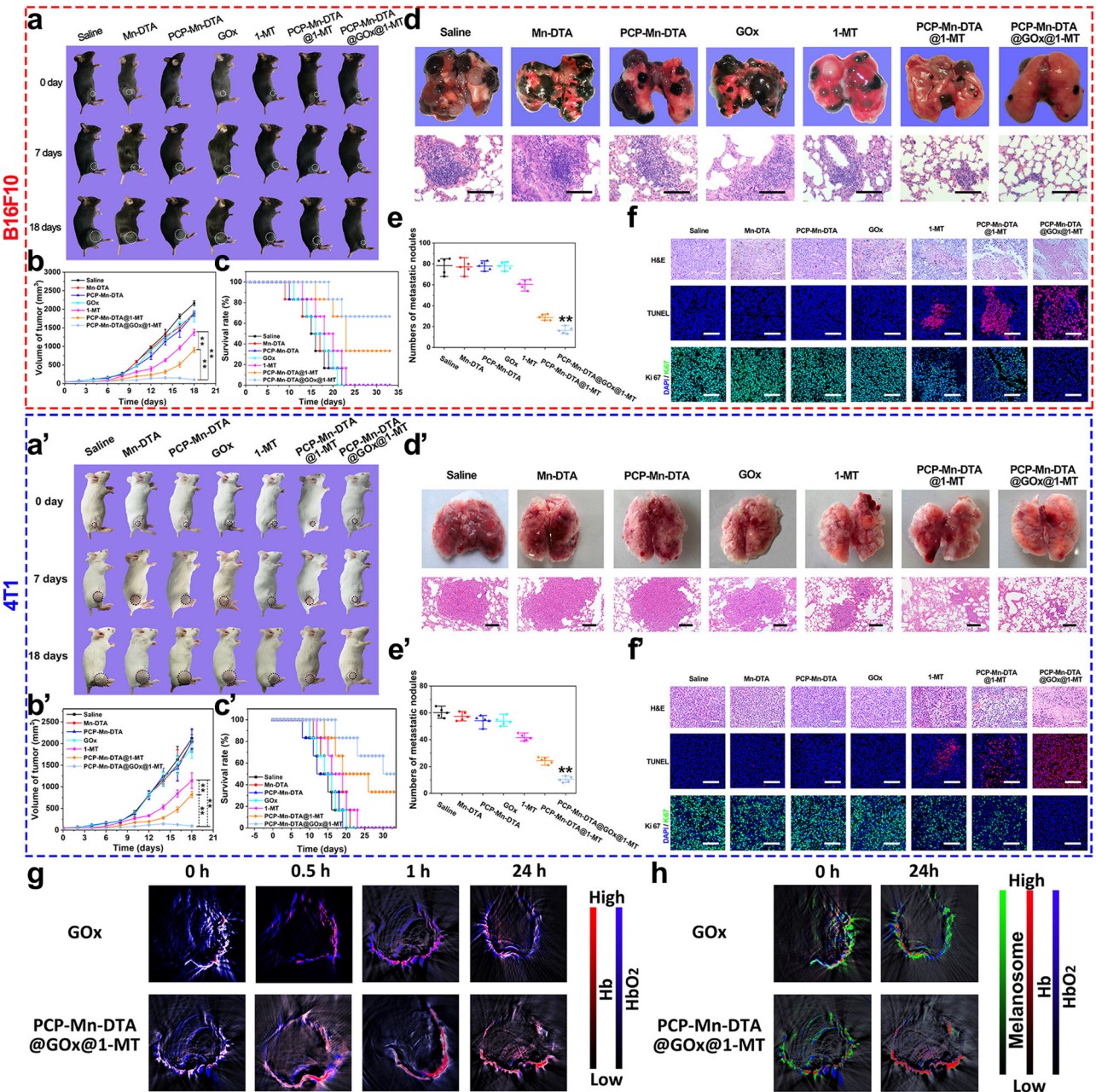

**Fig. 9 In vivo antitumor growth and metastasis as well as photoacoustic bioimaging of PCP-Mn-DTA@GOx@1-MT nanosystem. a, a'** Photographs of B16F10 (**a**) and 4T1 (**a'**) tumor-bearing mice after treatments with saline, Mn-DTA, PCP-Mn-DTA, GOx, 1-MT, PCP-Mn-DTA@1-MT, and PCP-Mn-DTA@GOx@1-MT for 0, 7, and 18 d, respectively. **b, b'** Tumor volumes and (**c, c'**) survival rates of mice after various treatments. **d, d'** Photographs of H&E staining, and (**e, e'**) quantitative analysis of the metastatic nodules from B16F10 and 4T1 metastatic lung tumor-bearing mice, respectively. **f, f'** TUNEL, H&E, and Ki67 staining images for tumors. **g** Photoacoustic images of oxygenated hemoglobin (HBO₂) and hemoglobin (HB), and (**h**) melanin signals in tumor sites in mice after intratumoral injection with GOx or PCP-Mn-DTA@GOx@1-MT for various time intervals, with images representative of 4 experiments. Scale bar: 100 μm for **d, d', f** and **f'**. Data represent mean ± SD (*n* = 6 biologically independent samples). *P*-values were determined by unpaired Student's *t*-test (two-tailed), **\*\*p** < 0.01.

was an insignificant change in the GOx negative group (Fig. 9h and Supplementary Fig. 22). These results indicated that the PCP-Mn-DTA@GOx@1-MT nanosystem effectively generated ROS and reduced melanin, thus, leading to a significantly enhanced tumor cell killing effect.

Furthermore, the biological safety of the nanosystem was studied. Firstly, the PCP-Mn-DTA@GOx@1-MT did not induce a continuous decrease in the peripheral blood glucose as compared to the control (Fig. 10a), suggesting its superior blood safety. Secondly, the blood biochemical levels as well as the hematological parameters (hematocrit value, mean platelet volume, hemoglobin, platelets, mean corpuscular hemoglobin, red blood cells, mean corpuscular hemoglobin concentration, red cell distribution width, mean corpuscular volume and white blood cells), liver function indices (aspartate aminotransferase and alanine aminotransferase) and kidney function indices (blood urea nitrogen and creatinine) remained unchanged after post-injection with PCP-Mn-DTA@GOx@1-MT with time (Supplementary Fig. 23), confirming a superior biocompatibility[49]. Thirdly, no obvious organ damage and body weight loss was

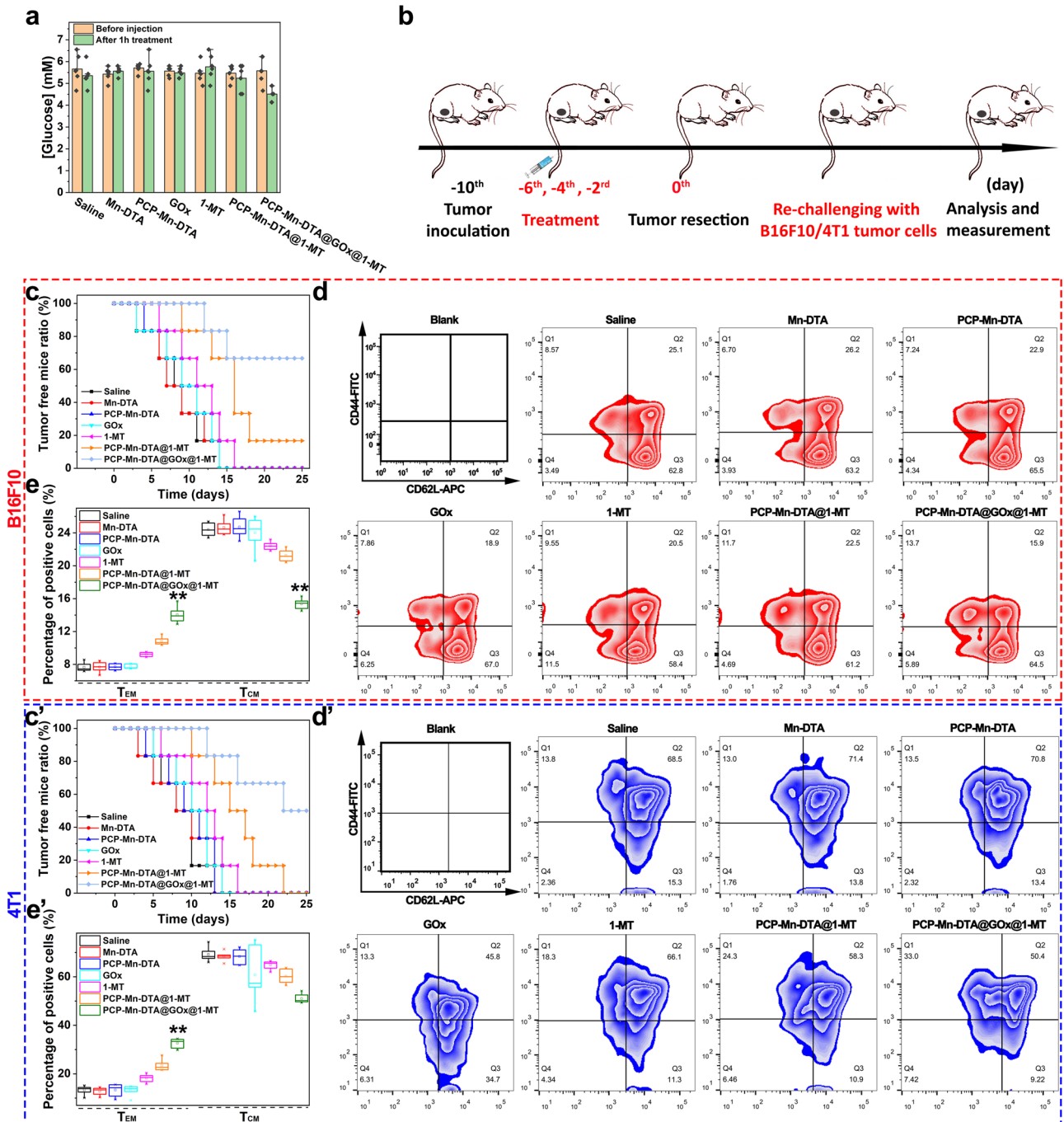

**Fig. 10 In vivo biosafety and antitumor recurrence effect of PCP-Mn-DTA@GOx@1-MT nanosystem. a** Changes in the blood glucose level after 1 h post-treatment. **b** Procedures of the treatment to study the recurrence degree of tumor. **c**, **c'** Tumor recurrence ratio of B16F10 and 4T1 tumor-bearing mice after various treatments. **d**, **d'** FCM and (**e**, **e'**) corresponding quantitative analysis on the frequency of $T_{EM}$ and $T_{CM}$ cells in the spleen at day 8 when re-challenging mice with secondary B16F10 and 4T1 tumors. Data represent mean ± SD ($n = 6$ biologically independent samples). *P*-values were determined by unpaired Student's *t*-test (two-tailed), **$p < 0.01$. Box plots indicate median (middle line), 25th and 75th percentile (box), and 5th and 95th percentile (whiskers), as well as outliers (single points).

observed in the case of PCP-Mn-DTA@GOx@1-MT group in both B16F10 and 4T1 tumor models, as revealed by the histological and H&E analysis (Supplementary Figs. 20, 24), thus, confirming the optimal biosafety in vivo. More importantly, PCP-Mn-DTA@GOx@1-MT also significantly prolonged the circulation time of 1-MT (Supplementary Fig. 25) and effectively accumulated 1-MT to the tumor site with a higher dose (nearly 5 times) as compared to free 1-MT. Furthermore, the amount of drug accumulation in the tumor was significantly higher than

other tissues ($p < 0.01$), attributed to the shielding effect and negative charge endowed by the PEG outer layer[50], as well as the reduced size and charge reversal design of the nanosystem.

Lastly, the tumor re-challenge and secondary response of the immune-memory T cell assays were conducted to systematically evaluate the persistence of the protective immune effect and potential clinic application of the nanosystem (Fig. 10b). As shown in Fig. 10c, c', PCP-Mn-DTA, 1-MT and PCP-Mn-DTA@1-MT demonstrated high tumor recurrence levels (83.3%-

100%), similar to the control group after 25 days of treatment. In comparison, PCP-Mn-DTA@GOx@1-MT exhibited a significant reduction in the recurrence ratio to 33.3% and 50% in the B16F10- and 4T1-based tumor re-challenge assays, respectively, suggesting an effective inhibition of the tumor recurrence. Moreover, PCP-Mn-DTA@GOx@1-MT treatment significantly increased the proportion of the effector memory T cells ($T_{EM}$) at day 8 (Figs. 10d, e and 10d', e') and 50 (Supplementary Fig. 26) under the 2nd tumor infusion, while the amount of the central memory T cells ($T_{CM}$) correspondingly decreased ($p < 0.01$). It confirmed the long term secondary immune response effect. The results indicated that the PCP-Mn-DTA@GOx@1-MT nanosystem could successfully initiate the immune memory response with a long-lasting effectiveness. Thus, the multifunctional PCP-Mn-DTA@GOx@1-MT nanosystem with the size reduction and charge conversion abilities could synergistically kill the tumor cells and suppress the tumor metastasis via combined advantages of starvation/oxidation-reinforced immune response initiation and IDO-blockade immune tolerance suppression.

## Discussion

In this study, a pH/ROS dual-responsive degradable MOF nanoreactor-based nanosystem has been developed with self-amplified release and enhanced penetration to co-deliver GOx and 1-MT for the combined starvation/oxidation therapy and IDO-blockade immunotherapy of tumors. The detailed in vitro and in vivo results have authenticated that the PCP-Mn-DTA@GOx@1-MT nanosystem not only strategically overcomes the biobarriers and improves the delivery efficiency via the weakly acidic tumor microenvironment-sensitive size/charge transition, but also effectively reinforces the immune response activation with reduced immune tolerance via the GOx-activated starvation/oxidation therapy and IDO-blockade immunotherapy. By taking advantage of these strategies, the PCP-Mn-DTA@GOx@1-MT nanosystem effectively suppresses the tumor growth and metastasis. Therefore, this study provides a promising paradigm for overcoming the biobarriers and reinforcing the immune response activation as well as relieving the immune resistance by the starvation/oxidation-integrated IDO-blockade immunotherapy.

## Methods

**Synthesis of ROS-responsive linker 5,5-dimethyl-4,6-dithia-nonanedioic acid (Compound 1)**. 3-Mercaptopropionic acid (4.9 mmol) was dissolved in acetone (9.82 mmol) and stirred for 8 h. Subsequently, the solution was crystallized in an ice bath overnight. The filtrated crystals were repeatedly washed by cold water and hexane, followed by drying under vacuum to obtain compound **1** (yield 81%). [1]H NMR (400 MHz, CD₃OD, ppm): 2.85 (t, 4H, -SCH₂CH₂-), 2.58 (t, 4H, -CH₂CH₂COOH), 1.58 (s, 6H, -SCCH₃CH₃S-). ESI-MS: Calc. 251.0417, found 251.0404. The [1]H NMR and mass spectra were presented in Fig. 2a, b.

**Synthesis of Mn-DTA nanoparticles**. The MnCl₂ solution (107 μL, 50 mg mL⁻¹ in DMF), compound **1** (347 μL, 15 mg mL⁻¹ in DMF), polyvinylpyrrolidone (K30, 300 mg) and triethylamine were added to a centrifuge tube. Afterwards, the DMF/ethanol mixture was added until the volume was 13 mL. Subsequently, the solution was transferred to a reactor and held at 150 °C for 24 h. Finally, the product, termed as Mn-DTA, was collected by centrifugation (18,500 g) and was re-dispersed in ethanol for further use.

**Synthesis of PEG-CDM (Compound 2)**. Briefly, Cis-aconitic anhydride (donated as CDM, 1 equiv.) and oxalyl chloride (2 equiv.) were dissolved into dry dichloromethane at 0 °C. Next, DMF was added dropwise and, the mixture was transferred to the room temperature condition for 2 h. The chloride acetylated CDM was harvested via vacuum drying and then reacted with mPEG-OH in the presence of pyridine for another 2 h. The saturated ammonium chloride was then added to terminate the reaction. The organic phase was extracted and precipitated by ice-cold ether twice. Compound **2** (PEG-CDM) was harvested via vacuum drying. [1]H NMR (400 MHz, CDCl₃, ppm): 7.95 (s, 1H, -CCHCOO-), 3.71 (m, 464H, -OCH₂CH₂-), 3.45 (s, 3H, -OCH₃) and 2.93 (s, 2H, -OOCH₂COO-). FTIR: 2885, 1726, 1633, 1569, 1471 and 1115 cm⁻¹. The [1]H NMR and FTIR spectra of **2** were shown in Supplementary Fig. 4a and Supplementary Fig. 5, respectively.

**Synthesis of PEG-CDM-PEI (Compound 3)**. Branched PEI (0.9 mmol) and compound **2** (0.6 mmol) were mixed with dimethyl sulfoxide (DMSO, 10 mL) at 0 °C. Afterwards, 4-dimethylaminopyridine (1.2 mmol) was added dropwise and stirred for 0.5 h. The reaction contents were warmed to room temperature in the dark, and the reaction was continued for 2 h. The mixture was then dialyzed (MWCO 3.5 kDa) against distilled water for 3 days. Compound **3**, also denoted as PEG-CDM-PEI, was achieved by the lyophilization (yield: 82%). [1]H NMR (400 MHz, D₂O, ppm): 3.83 (m, 464H, -OCH₂CH₂-), 3.31 (s, 3H, -OCH₃) and 2.62-2.97 (t, -CH₂CH₂N-). The molecular weight was determined to be 7200 by GPC, with a polydispersity index of 1.29. FTIR: 2885, 1739, 1645, 1562, 1471 and 1116 cm⁻¹. The [1]H NMR, GPC and FTIR spectra of **3** were shown in Supplementary Fig. 4b, c and Supplementary Fig. 5, respectively.

**Preparation of PEG-CDM-PEI-Mn-DTA nanoparticles**. Mn-DTA (10 mg), 1-ethyl-3-(3-dimethyl amino propyl) carbodiimide hydrochloride (EDC·HCl, 6 mM) and N-hydroxysuccinimide (NHS, 6 mM) were dissolved in PBS (pH 7.4, 10 mL). After stirring for 1.5 h, compound **3** (0.06 mmol) was added to the solution, which was reacted for another 36 h. The product, denoted as PCP-Mn-DTA, was collected by centrifugation (18,500 g) and vacuum drying.

**Preparation of GOx and 1-MT co-loaded nanoparticles**. Mn-DTA (10 mg), GOx (1.33 mg) and 1-MT (2 mg) were dissolved in PBS (pH 7.4). After stirring at room temperature for 24 h, the mixture was centrifuged (18,500 g), followed by resuspension in PBS (pH 6.0, 10 mL) containing EDC·HCl (15 mM) and NHS (15 mM). The conjugation of compound **3** was conducted similar to the synthesis of the PCP-Mn-DTA nanoparticles. Finally, the drug-loaded PCP-Mn-DTA@GOx@1-MT was harvested by centrifugation (18,500 g) and lyophilization.

The loading of GOx and 1-MT in PCP-Mn-DTA@GOx@1-MT was quantified by UV-vis spectroscopy based on the standard curves of GOx at 276 nm and 1-MT at 288 nm, respectively. The drug loading content (DLC) and drug loading efficiency (DLE) were calculated by following equations:

$$DLC\,(\%) = \frac{\text{Amount of loaded drug}}{\text{Weight of nanosystem}} \times 100\% \qquad (1)$$

$$DLE\,(\%) = \frac{\text{Amount of loaded drug}}{\text{Weight of drug in feed}} \times 100\% \qquad (2)$$

**Detection of glucose oxidase activity**. In order to detect the activity of GOx loaded in the PCP-Mn-DTA@GOx@1-MT nanosystem, glucose (1 mg/mL) was mixed with free GOx (200 μg/mL) or an equivalent amount of GOx loaded PCP-Mn-DTA@GOx@1-MT (2.27 mg/mL). The supernatant was taken out at different time intervals (0 h, 0.1 h, 0.5 h, 1 h, 1.5 h and 2 h), and the H₂O₂ generation and pH change were monitored.

**Drug release behavior**. PCP-Mn-DTA@GOx@1-MT (3 mg) was first dissolved in PBS (1 mL, pH 7.4) without or with different concentrations of H₂O₂ or glucose, followed by transferring to the dialysis bags. The dialysis bags were subsequently immersed in PBS (29 mL) with different conditions under stirring (100 rpm) at 37 °C. The medium (0.7 mL) was taken out at desired time intervals (1 h, 2 h, 3 h, 4 h, 5 h, 12 h, 24 h, and 48 h), and the bags were refreshed with the same volume of PBS. The released GOx was measured by UV-vis spectroscopy at a wavelength of 276 nm.

**Stability assay of nanosystem in serum**. Briefly, PCP-Mn-DTA@GOx@1-MT (1 mg/mL) was incubated with 10% fetal bovine serum (FBS) in PBS (pH 7.4 and 6.8) at 37 °C for 6 days. The sample solution was taken out at specific time intervals (0 h, 6 h, 18 h, 24 h, 48 h, 72 h, 96 h, 120 h and 144 h) and was detected by DLS to analyze the changes in the size of PCP-Mn-DTA@GOx@1-MT.

**Cell culture and cytotoxicity assay**. B16F10 melanoma cells were purchased from Cell Bank of Chinese Academy of Sciences, China. The cells were cultivated in Dulbecco's modified Eagle's medium (DMEM) supplemented with 1% (w/v) penicillin (100 U/mL)/streptomycin (100 μg/mL) and 10% (v/v) FBS having 5% CO₂ at 37 °C.

To evaluate the cytotoxicity, the B16F10 cells (2 × 10⁴ cells per square centimeter) seeded on the 24-well plates were treated with PBS, 1-MT (7 μM), PCP-Mn-DTA (11.36 μg/mL), PCP-Mn-DTA@1-MT, PCP-Mn-DTA@GOx@1-MT (11.36 μg/mL, equivalent of 10 mU/mL GOx and 7 μM 1-MT), and PCP-Mn-DTA@GOx@1-MT + glucose (5.5 mM glucose) for 24 h and 48 h, followed by incubating with the mixture solution composed of 200 μL fresh medium and 20 μL CCK-8 at 37 °C for another 1.5 h. The absorbance at 450 nm was recorded by a spectrophotometric microplate reader (Bio-Rad 680, USA).

**In vitro cellular uptake**. The endocytosis of the B16F10 cells on the PCP-Mn-DTA nanosystem was observed by CLSM and quantitatively analyzed by FCM. On one hand, the B16F10 cells seeded on the confocal microscope dish were treated with PBS, FITC-labelled GOx (GOx-FITC, 10 mU/mL) and PCP-Mn-DTA@GOx-

FITC@1-MT at pH 7.4 and 6.8 for 4 and 12 h. The cells were subsequently fixed with 4% paraformaldehyde, permeabilized with 0.5% TritonX-100, followed by staining with ActinRed 555 and 4',6-diamidino-2-phenylindole (DAPI). CLSM (LSM 510 META Olympus) and CellSens Dimension software were employed to study the efficiency of uptake by the B16F10 cells. On the other hand, the B16F10 cells treated with the various systems were washed with PBS. The cells were subsequently centrifuged (800 g × 10 min) at 4 °C and collected. Finally, they were resuspended in the cell binding solution (300 μL) and analyzed by FCM (FACS Calibur and Celesta, BD, Biosciences) and FlowJo_V10 software.

**Measurement of intracellular ROS generation.** Typically, DCF-DA was used as the ROS probe to monitor the ROS generation in the B16F10 cells. Briefly, the B16F10 cells seeded on the 6-well plates or confocal microscope dishes were treated with GOx, PCP-Mn-DTA, PCP-Mn-DTA@GOx, PCP-Mn-DTA@GOx@1-MT and PCP-Mn-DTA@GOx@1-MT + glucose with the same concentration of GOx. The cellular ROS generation was imaged by CLSM and quantitatively analyzed by FCM.

**MCS construction and tumor penetration assay.** Firstly, hot agarose (1.5%) solution was poured into a 96-well plate (80 μL per well), which was cooled to form a layer of agaropectin. Next, MCSs were formed after incubating B16F10 cells on this plate about a week[19]. Next, MCSs were transferred to two 6-well plates containing DMEM with different pH values, followed by treatment with the PCP-Mn-DTA@GOx@1-MT nanoparticles (11.36 μg/mL) labelled with FITC for 4 and 12 h. The resultant MSCs were washed by PBS, re-suspended in fresh medium and subsequently detected by CLSM.

**In vitro IDO activity inhibition.** Briefly, IDO1 (IDO1 gene was transfected with Lipofectamine®3000, NCBI gene)-transfected B16F10 positive cells and B16F10 cells were incubated with PBS, PCP-Mn-DTA, 1-MT, PCP-Mn-DTA@1-MT, and PCP-Mn-DTA@GOx@1-MT for 24 h, respectively. Next, the cells were lysed and obtained by centrifugation (800 g × 10 min, 4 °C). The western blotting analysis was used to measure the expression of IDO1.

**In vitro T-cell proliferation assay.** IFN-γ-stimulated B16F10 cells (1 × 10⁵ cells/well) were mixed with the T cells (5 × 10⁵ cells/well) in a 24-well plate, followed by treating with 1-MT, PCP-Mn-DTA, PCP-Mn-DTA@1-MT and PCP-Mn-DTA@-GOx@1-MT, respectively. After 12 h co-culturing, the lymphocytes were obtained and incubated with EdU (10 μM) for the FCM analysis.

**Lysosome escaping analysis.** After the confluency of the B16F10 cells seeded on the confocal microscope dishes reached 60-70%, the cells were incubated with FITC labelled PCP-Mn-DTA@GOx@1-MT (11.36 μg/mL) and Lysotracker red (1 μM) at 37 °C for 1, 3, and 8 h. Afterwards, the cells were then washed with PBS and observed by CLSM.

**FCM analysis of cell apoptosis.** Once the confluence of the B16F10 cells seeded in a 6-well plate reached 60–70%, the cells were treated with PBS (Control), 1-MT (7 μM), PCP-Mn-DTA (11.36 μg/mL), PCP-Mn-DTA@1-MT, PCP-Mn-DTA@-GOx@1-MT (11.36 μg/mL, equivalent of 7 μM 1-MT) and PCP-Mn-DTA@-GOx@1-MT + glucose (5.5 mM glucose) for 24 h. The cells were subsequently obtained by centrifugation (800 g), and stained by the Annexin V-FITC/PI kit (NeoBioscience) according to the instructions, followed by the FCM analysis.

**Western-blot analysis.** B16F10 cells seeded on the 6-well plates were treated with PBS, 1-MT (7 μM), PCP-Mn-DTA, PCP-Mn-DTA@1-MT, PCP-Mn-DTA@-GOx@1-MT (11.36 μg/mL, equivalent of 7 μM 1-MT) and PCP-Mn-DTA@-GOx@1-MT + glucose for 24 h, respectively. Afterwards, the cells were lysed, and the protein samples were collected through centrifugation (18,500 g). Western blotting (Gel DocTM XR + , Bio-Rad) and ImageJ software 1.45 f were used to monitor and analyze the expression of Bax, Bcl-2 and Cyt-C (Supplementary Fig. 27).

**Wound healing assay.** The B16F10 cells were seeded in the 6-well plates and incubated for 24 h. After treatment with PBS, PCP-Mn-DTA, 1-MT, PCP-Mn-DTA@1-MT and PCP-Mn-DTA@GOx@1-MT for 24 h, a scratched wound was produced by wounding the confluent cell monolayers with a p200 pipette tip. Afterwards, the cells were washed to remove the nanosystem. After 24 h, the images of wound healing were acquired using a microscope (Olympus).

**In vitro migration and invasion.** The B16F10 cells were treated with the different systems used herein for 24 h. Later, the cells were digested and collected by centrifugation (800 g). For the cell migration experiment, the cells were resuspended in a serum-free medium and subsequently transferred to the upper chamber of Transwell at a density of 1 × 10⁵ cells. For the invasion experiment, 2 × 10⁵ B16F10 cells were cultured on the Transwell pre-covered matrix gel. Meanwhile, a medium containing 10% FBS was added to the lower chamber as a chemical attractant. After

incubation for 24 h, the cells on the lower surface of Transwell were stained by crystal violet. They were subsequently counted and imaged by using a microscope.

**In vivo antitumor efficacy.** C57BL/6 mice and Balb/c mice (about 6 weeks old) were purchased from Beijing Institution for Drug Control (China). All animals were bred in the pathogen-free facility with a 12 h light/dark cycle and relative humidity (40–70%) at 21 ± 2 °C. All the mice had access to food and water ad libitum. The animal studies were conducted according to the guidelines of the Institutional Animal Care and Use Committee at Northwestern Polytechnical University. The tumor-bearing mouse models were established by inoculating B16F10 or 4T1 cells (100 μL, concentration of 1 × 10⁶) in PBS into the right flank of the C57BL/6 or Balb/c mice[19]. Once the tumor volume reached about 50 mm³, seven groups of the tumor-bearing mice were intravenously injected with saline, Mn-DTA, PCP-Mn-DTA, GOx, 1-MT, PCP-Mn-DTA@1-MT and PCP-Mn-DTA@GOx@1-MT at a dose of 3 mg/kg 1-MT (n = 6). These treatments were operated twice per week and lasted for 18 days. The tumor volumes and body weights of mice were measured every 2 days. The tumor volume (V) was calculated by using the equation:

$$V = \frac{L \times S^2}{2} \tag{3}$$

where L is the longest dimension of tumor, and S is the shortest dimension of tumor. After the last treatment, the mouse survival rate was recorded for another 15 days.

**Quantification of tumor-infiltrating lymphocytes.** The B16F10 and 4T1 cell-bearing mice were intravenously injected with the same seven formulations. The injections were operated twice every 2 days. Subsequently, the tumor tissues were excised, digested and harvested into mono-dispersive lymphocytes. The obtained lymphocytes were co-stained with anti-CD4-FITC, anti-CD3-APC/Cy7, and anti-CD8-Percp/Cy5.5 antibodies for analyzing the CD4⁺ (CD3⁺CD4⁺CD8⁻) and CD8⁺ T cells (CD3⁺CD4⁻CD8⁺), co-marked with anti-CD3-APC/Cy7 and anti-CD49b-PE/Cy7 for analyzing the NK cells (CD3⁻CD49b⁺), co-marked with anti-CD3-APC/Cy7 and anti-CD45R/B220-PE for analyzing the B cells (CD3⁻CD45R/B220⁺), co-marked with anti-CD80-APC, anti-CD86-PE and anti-CD11b-FITC for analyzing the mature dendritic cells (CD11b⁺CD80⁺CD86⁺), and co-marked with anti-CD4-FITC, anti-Foxp3-Alexa 647, and anti-CD3-APC/Cy7 for the analysis of the Treg cells by using FCM, according to the procedures of the manufacturer.

**Quantification of memory T cells with FCM.** The splenocytes collected from the treated mice were marked with anti-CD44-FITC, anti-CD3-PerCP-Cy5.5, anti-CD62L-APC, and anti-CD8-PE antibodies according to the procedures of the manufacturer. Afterwards, the detection with FCM was carried out to measure the central and effector memory T cells.

**Tryptophan and kynurenine measurement assay in vivo.** The B16F10 cell-bearing mice were intravenously injected with the formulations every 3 days for 5 times. The tumor samples were harvested, homogenized, centrifuged and measured by using high performance liquid chromatography-mass spectrometry (HPLC-MS).

**Photoacoustic imaging.** Tumor-bearing mice were intravenously injected GOx or PCP-Mn-DTA@GOx@1-MT for various time intervals. The photoacoustic signals of oxygenated hemoglobin, hemoglobin and melanin in tumor sites were detected with iThera Medical MSOT inVision 128 (iThera Medical) and analyzed with ViewMOST (Release 3.8.1.09) software.

**H&E, TUNEL, and IFC studies.** The mice were sacrificed after the treatment for 18 days. The spleen, kidney, lung, heart, liver, and tumor were resected and made into sections. For the H&E assay, the sections of the tissues were stained with H&E and imaged by using a microscope (BX53, Olympus) with CellSens Entry software. For the TUNEL assay, the tumor sections were stained with the TUNEL agents (Beyotime) and imaged by a CLSM. For the IFC study, the tumor sections were blocked by 5% bovine serum albumin and incubated with Ki67, IDO, mTOR, or STAT3 antibody. Afterwards, they were stained with Alexa Fluor 488-labeled IgG or Cy3-labeled second antibody according to the manufacturer's instructions (Immunol Fluorescence Staining Kit, Beyotime, China). Finally, the sections were stained with DAPI and characterized using a CLSM.

**Determination of blood glucose concentration in mice, blood safety and liver/ kidney toxicity assay.** PCP-Mn-DTA@GOx@1-MT along with control (saline) were intravenously injected into the mice. After 1 h of administration, the blood was collected for the blood glucose determination. The blood glucose level of the mice was measured by using a blood glucose meter before and after the administration. Additionally, the blood of the mice was collected directly from the eyes after the last treatment. The blood samples were stored at 4 °C overnight and centrifuged at 900 g for 20 min to separate the plasma. Subsequently, the blood

biochemical levels and hematological parameters as well as the liver and kidney function indices were determined according to the protocols recommended by the manufacturers.

**Pharmacokinetic and biodistribution study**. The B16F10 tumor-bearing C57BL/6 mice were intravenously injected with free 1-MT and PCP-Mn-DTA@GOx@1-MT at an equivalent 1-MT dose of 3 mg/kg. For pharmacokinetic study, at the desired time intervals, the blood was collected from the orbital plexus and heparinized. The plasma was harvested by centrifugation (2000 g). Acetonitrile was then added to precipitate proteins, followed by centrifugation at 18,500 g for 8 min to collect the supernatant. Finally, the collected sample was dried, re-dissolved and detected by HPLC. For biodistribution study, the mice were sacrificed after 24 h, and the tumor and major organs were collected and homogenized in DMSO (0.5 mL), followed by centrifugation at 16,000 g for 15 min. The amount of 1-MT in these organs was determined by HPLC.

**Tumor re-challenge assay**. After primary tumor inoculation, the B16F10-bearing C57BL/6 mice and 4T1-bearing Balb/c mice ($n = 6$) were intravenously injected with saline (control), Mn-DTA, PCP-Mn-DTA, GOx, 1-MT, PCP-Mn-DTA@1-MT and PCP-Mn-DTA@GOx@1-MT on day -6, -4 and -2. The primary tumors were surgically resected on day 0, and the mice were subcutaneously re-challenged with the 4T1 or B16F10 cells and monitored for the recurrence of the secondary tumors.

**Statistical analysis**. The statistical analysis was performed using the OriginPro software (version 9.0 and 2022) by the Student's t-test and one-way analysis of variance (ANOVA). The data were expressed as means ± standard deviation (SD). The confidence levels of 95% and 99% were regarded as the significant difference.

**Reporting summary**. Further information on research design is available in the Nature Research Reporting Summary linked to this article.

## Data availability

All the data supporting the findings of this study are available within the article and its supplementary information files and from the corresponding author upon reasonable request.

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

## Acknowledgements

This work was financially supported by the National Natural Science Foundation of China (52003223 (L.D.), 11722220 (H.Y.), and 11672246 (H.Y.)), the National Key R&D Program of China (2016YFC1100300 (K.C.)), the Key R & D Program of Shaanxi Province (2022SF-012 (L.D.)), and the Young Talent fund of University Association for Science and Technology in Shaanxi (20200302 (L.D.)). This work was partially supported by the Singapore Agency for Science, Technology and Research (A*STAR) AME IRG grant (A20E5c0081 (Y.Z.)), and the Singapore National Research Foundation Investigatorship (NRF-NRFI2018-03 (Y.Z.)).

## Author contributions

L.D. conceived the project. L.D. and M.Y. designed the experiments and analyzed the results. M.Y. and S.M. performed confocal microscopy imaging. M.Y., Z.F., and X.L. performed western blot assays and analysis. M.Y., Z.F., and X.Z. performed the in vivo studies. Z.Y. assisted histological and immunofluorescence analysis. K.C., H.Y., and Y.Z. provided the discussions. The paper was written by L.D., M.Y., and Y.Z. All authors contributed to the general discussions and reviewed the paper.

## Competing interests

The authors declare no competing interests.
