## [Peer Review File · Nature Communications]

Multifunctional metal-organic framework-based nanoreactor for starvation/oxidation improved indoleamine 2,3-dioxygenase-blockade tumor immunotherapyREVIEWER COMMENTS

Reviewer #1 (Remarks to the Author):

Glucose oxidase (GOx) and indoleamine 2,3-dioxygenase (IDO) inhibitor 1-methyltryptophan (1-MT) are co-loaded with metal-organic framework (MOF)-based multifunctional nanoreactor. The authors studied this nanoreactor for starvation/oxidation improved indoleamine 2,3-dioxygenase-blockade tumor immunotherapy. This manuscript does not show enough advantage over previous GOx and IDO works.

- 1 Please highlight the novelty by comparing with previous works.
- 2 MOF is inorganic materials and it might bring long-term toxicity over other nanoreactors. What is the advantage of MOF over other nanoreactors?
- 3 For in vivo experiments, the MOF and GOx are better to be used as control.
- 4 SEM images are better to be supplied.
- 5 Can the authors characterize the MOF with HRTEM to understand the detailed structure?

Reviewer #2 (Remarks to the Author):

In this paper, the authors reported a pH/ROS dual-sensitive degradable MOF nanoreactor-based nanosystem with self-amplified drug release and enhanced tumor penetration to co-deliver GOx and 1-MT for tumor starvation/oxidation/IDO-blockade immunotherapy. Most of the characterization and experimental designs were thorough and results were presented logically. However, the efficacy of this treatment was only shown in one cancer model. My main concern is the applicability of these results to other cancer models and clinical translation of these results. This is the most critical issue and the authors must prove efficacy in additional (and more clinically relevant tumor models) for the paper to be suitable for publication in Nature Communications.

Reviewer #3 (Remarks to the Author):

In this manuscript, the authors reported a pH/ROS dual-responsive degradable MOF nanoreactor-based system, co-delivering GOx and 1-MT, with self-amplified release and enhanced penetration for combined starvation/oxidation therapy and IDO-blockade immunotherapy of tumor. The as-prepared PCP-Mn-DTA@GOx@1-MT nanosystem could strategically improve the delivery efficiency and reinforce immune response activation via the weakly acidic tumor microenvironment-sensitive size/charge transition, GOx-activated starvation/oxidation therapy and IDO-blockade immunotherapy strategies. The results are interesting and could be published in Nature Communications, however, some points need the authors to be addressed before publication as follows:

1. As for drug delivery, the toxicity of nanocarriers is essential. The cell viability of PCP-Mn-DTA nanocarrier with different dosages should be measured.
2. The first-time application of GOx for in vivo cancer starvation therapy is the Reference 25, not 3.
3. This point should be discussed in the introduction.
3. Since GOx could consume glucose and inhibit ATP production by starvation treatment, the level of intracellular ATP upon above treatments thus should be investigated.
4. Please add the live-dead assay of tumor cells in vitro upon different treatments to comprehensively evaluate tumor killing effect mediated by nanosystem.
5. In the section of in vivo biosafety study, although the authors investigated the changes of mouse body weight, major organ pathological analysis, and liver function evaluations, more biosafety experiments need to be provided, for example, the biosafety of kidney, blood compatibility, etc.
6. Please add the detailed procedures about tumor re-challenge assay.
7. Some recent works on GOx-instructed synergistic cancer therapy should be cited.

Responses to Reviewers' Comments

Major changes made based on reviewers' comments:

- (1) We preformed the animal studies again and added MOF and GOx treatments as controls based on the suggestions from the reviewer #1. Please see **Figure 8-10** in the revised manuscript, and **Supplementary Fig. 13-18** and **Supplementary Fig. 22** in the supporting information.
- (2) We provided the SEM image of MOF based on the suggestions from the reviewer #1. Please see **Supplementary Fig. 1a** in the supporting information.
- (3) We characterized the MOF with HRTEM, described its detailed structure and added the related discussions based on the suggestions from the reviewer #3. Please see **Supplementary Fig. 1b** in the revised supporting information, and **page 6** in the revised manuscript, as marked with YELLOW backgrounds.
- (4) We additionally performed another tumor model (4T1 tumor cells-bearing Balb/c mouse model) and carefully studied the *in vivo* immune response level, and *in vivo* antitumor effect of the nanosystem based on the suggestions from the reviewer #2. Please see **Figure 8-10** in the revised manuscript, and **Supplementary Fig. 13-18** and **Supplementary Fig. 22** in the supporting information.
- (5) We measured the cell viability of PCP-Mn-DTA nanocarrier with different dosages against B16F10 cells based on the suggestions from the reviewer #3. Please see **Supplementary Fig. 8** in the supporting information.
- (6) We provided the biosafety of kidney and blood compatibility assays, including the blood biochemical levels, hematological parameters and kidney function indexes based on the suggestions from the reviewer #1 and reviewer #3. Please see **Supplementary Fig. 21** in the supporting information, and **page 20** in the revised manuscript, as marked with YELLOW backgrounds.
- (7) We added the live-dead assay of tumor cells in vitro upon different treatments based on the suggestions from the reviewer #3. Please see **Supplementary Fig. 11** in the supporting information.
- (8) We added the level of intracellular ATP upon above treatments based on the suggestions from the reviewer #3. Please see **Supplementary Fig. 10** in the revised supporting information.
- (9) We justified the cited reference about the first-time application of GOx for in vivo cancer starvation therapy in the introduction based on the suggestions from the reviewer #3. Please see **page 2** and **reference 2** in the revised manuscript.
- (10) We added the detailed procedures and schematic illustration about tumor re-challenge assay based on the suggestions from the reviewer #3. Please see **page 31** and **Figure 10b**

in the revised manuscript.

- (11) We cited some recent papers in GOx-instructed synergistic cancer therapy based on the suggestions from the reviewer #3. Please see **references 2-4** in the revised manuscript.
- (12) We further highlighted the novelty by comparing with previous works and cited the related references based on the suggestions from the reviewer #1 in the revised manuscript. Please see **page 4** and **page 5**, as marked with YELLOW background.

Reviewer #1

C: Glucose oxidase (GOx) and indoleamine 2,3-dioxygenase (IDO) inhibitor 1-methyltryptophan (1-MT) are co-loaded with metal-organic framework (MOF)-based multifunctional nanoreactor. The authors studied this nanoreactor for starvation/oxidation improved indoleamine 2,3-dioxygenase-blockade tumor immunotherapy. This manuscript does not show enough advantage over previous Gox and IDO works.

A: Thanks for your criticism with instructive questions. Based on your questions, we revised the manuscript one by one as follows.

Q1. *Please highlight the novelty by comparing with previous works.*

A: Thanks for your good suggestions. We highlighted the novelty by comparing with previous works and cited the related references in the revised manuscript according to your suggestion. Please see **page 4** and **page 5** in the revised manuscript. All changes were marked with YELLOW backgrounds.

Q2. *MOF is inorganic materials and it might bring long-term toxicity over other nanoreactors. What is the advantage of MOF over other nanoreactors?*

A: Thanks very much for your kind reminding. On the one hand, compared to other nanoreactors, MOFs have the advantages of facile preparation, high loading efficacy of enzymes, excellent fidelity characteristics of enzyme activity, multifunctionality and good biocompatibility, making them promising systems for developing the nanoreactors [1-4]. On the other hand, tumor-activated degradable MOF nanoreactor was synthesized for the first time in this work via covalent crosslinking with ROS-susceptible agents and Mn^{2+} , which can be rapidly disassembled and excreted from body in response to the rich intracellular ROS of tumor cells over other nanoreactors, minimizing the potential long-term retention toxicity of conventional nanoreactors.

More importantly, we further investigated the long-term toxicity of MOF in this work. Firstly, we monitored the time- and dose-dependent cell cytotoxicity. The result demonstrated that B16F10 cells treated with various concentrations of MOF-based nanoreactor (PCP-Mn-DAT) for 12 h and 24 h all exhibited good cell viability similar to that of the control (**Supplementary Fig. 8**). Next, as shown in **Supplementary Fig. 21**, the blood biochemical levels and hematological parameters (hematocrit value (HCT), mean platelet volume (MPV), hemoglobin (HGB), platelets (PLT), mean corpuscular hemoglobin (MCH), red blood cells (RBC), mean corpuscular hemoglobin concentration (MCHC), red cell distribution width (RDW), mean corpuscular volume (MCV) and white blood cell (WBC)), liver function indexes (aspartate transaminase (AST) and alanine aminotransferase (ALT)), and kidney function indexes (blood urea nitrogen (BUN) and creatinine (CREA)) of the mice did not show obvious changes after post-injection with PCP-Mn-DTA@GOx@1-MT for long-term period (18 days post-injection), indicating that

there was no significant long-term toxicity during the evaluation period. Furthermore, there was also no evident difference on the body weight and histological analysis of the major organs of two tumor models (B16F10 tumor-bearing C57BL/6 mice and 4T1 tumor-bearing Balb/c mice) after the i.v. injection with MOF nanoreactor-based nanosystem over 18 days (**Supplementary Fig. 18** and **Supplementary Fig. 22**), confirming again the good biocompatibility. Above primary and comprehensive *in vitro* and *in vivo* results indicated that MOF nanoreactor-based nanosystem showed no obvious toxicity, which provided a chance for *in vivo* therapeutic applications. The related descriptions have been added in the revised manuscript. Please see **Supplementary Fig. 8**, **Supplementary Fig. 18**, **Supplementary Fig. 21** and **Supplementary Fig. 22** in the supporting information, **page 4** and **page 20** in the revised manuscript, as marked with YELLOW backgrounds.

References:

- [1] Lian, X.Z., Huang, Y. Y., Zhu, Y. Y., Fang, Y., Zhao, R., Joseph, E., Li, J. L., Pellois, J.P. & Zhou, H. C. Enzyme-MOF nanoreactor activates nontoxic paracetamol for cancer therapy. *Angew. Chem. Int. Ed.* **13**, 5827-5832 (2018).
- [2] Song, G. S., Chen, Y. Y., Liang, C., Yi, X., Liu, J. J., Sun, X. Q., Shen, S.D., Yang, K. & Liu, Z. Catalase-loaded TaOx nanoshells as bio-nanoreactors combining high-z element and enzyme delivery for enhancing radiotherapy. *Adv. Mater.* **23**, 7143-7148 (2016).
- [3] Wu, M. X. & Yang, Y. W. Metal-organic framework (MOF)-based drug/cargo delivery and cancer therapy. *Adv. Mater.* **29**, 1606134 (2017).
- [4] Zhang, L., Wang, Z. Z., Zhang, Y., Cao, F. F., Dong, K., Ren, J. S. & Qu, X. G. Erythrocyte membrane cloaked metal-organic framework nanoparticle as biomimetic nanoreactor for starvation-activated colon cancer therapy. *ACS Nano* **14**, 10201-10211 (2018).

Q3. For *in vivo* experiments, the MOF and GOx are better to be used as control.

A: Thanks for your kind suggestions. We preformed the animal studies again and added MOF and GOx treatments as controls, according to your suggestion. Please see section of “**In vivo immune response reinforcement and immune evasion inhibition of PCP-Mn-DTA@GOx@1-MT**” and “**In vivo antitumor effect of PCP-Mn-DTA@GOx@1-MT**”, **Figure 8-10** in the revised manuscript, **Supplementary Fig. 13-18** and **Supplementary Fig. 22** in the supporting information.

Q4. SEM images are better to be supplied.

A: Thanks for your kind suggestions. We provided the SEM image of MOF in the revised supporting information. Please see **Supplementary Fig. 1a** in the supporting information.

Q5. Can the authors characterize the MOF with HRTEM to understand the detailed structure?

A: Thanks for your suggestion. We characterized the MOF with HRTEM, described its detailed

structure and added the related discussions, according to your suggestion. As shown in **Supplementary Fig. 1b**, Mn-DTA with a clear structure exhibits lattice spacings of 0.305 and 0.328 nm detected by HRTEM, which are consistent with values for the (101, 002) planes of MnS as determined by powder XRD. Please see **Supplementary Fig. 1b** in the supporting information, **page 6** in the revised manuscript, as marked with YELLOW backgrounds.

Reviewer #2

C: In this paper, the authors reported a pH/ROS dual-sensitive degradable MOF nanoreactor-based nanosystem with self-amplified drug release and enhanced tumor penetration to co-deliver GOx and 1-MT for tumor starvation/oxidation/IDO-blockade immunotherapy. Most of the characterization and experimental designs were thorough and results were presented logically. However, the efficacy of this treatment was only shown in one cancer model. My main concern is the applicability of these results to other cancer models and clinical translation of these results. This is the most critical issue and the authors must prove efficacy in additional (and more clinically relevant tumor models) for the paper to be suitable for publication in Nature Communications.

A: Thanks for your criticism with instructive questions. Based on your suggestions, we additionally selected another relevant tumor model (4T1 tumor cell-bearing Balb/c mouse model) and carefully studied the *in vivo* immune response level and *in vivo* antitumor effect of the nanosystem (**Figure 8-10, Supplementary Fig. 13-18 and Supplementary Fig. 22**), similar to that on B16F10-bearing C57 mouse model. Both two tumor model results primarily confirmed that MOF nanoreactor-based nanosystem exhibited good antitumor efficiency with reinforced immune response activation and relieved immune resistance, which provided a common chance for *in vivo* therapeutic applications. Please see section of “**In vivo immune response reinforcement and immune evasion inhibition of PCP-Mn-DTA@GOx@1-MT**” and “**In vivo antitumor effect of PCP-Mn-DTA@GOx@1-MT**”, **Figure 8-10** in the revised manuscript, **Supplementary Fig. 13-18 and Supplementary Fig. 22** in the supporting information.

Reviewer #3

C: In this manuscript, the authors reported a pH/ROS dual-responsive degradable MOF nanoreactor-based system, co-delivering GOx and 1-MT, with self-amplified release and enhanced penetration for combined starvation/oxidation therapy and IDO-blockade immunotherapy of tumor. The as-prepared PCP-Mn-DTA@GOx@1-MT nanosystem could strategically improve the delivery efficiency and reinforce immune response activation via the weakly acidic tumor microenvironment-sensitive size/charge transition, GOx-activated starvation/oxidation therapy and IDO-blockade immunotherapy strategies. The results are interesting and could be published in Nature Communications, however, some points need the authors to be addressed before publication as follows:

A: Thanks for your appreciation of our work with instructive suggestions. Based on your suggestions, we revised the manuscript one by one as follows.

Q1. *As for drug delivery, the toxicity of nanocarriers is essential. The cell viability of PCP-Mn-DTA nanocarrier with different dosages should be measured.*

A: Thanks so much for your kind suggestion. We studied the cell viability of PCP-Mn-DTA nanocarrier with different dosages against B16F10 cells according to your suggestion. This research demonstrated that the cell viability of PCP-Mn-DTA with a serial of dosages (50-1000 µg/mL) had no obvious difference from control group after 12 h or 24 h incubation (**Supplementary Fig. 8**), confirmed again the low toxicity of PCP-Mn-DTA nanosystem. Please see **page 9** and **page 10** in the revised manuscript, and **Supplementary Fig. 8** in the supporting information. All changes were marked out with YELLOW backgrounds.

Q2. *The first-time application of GOx for in vivo cancer starvation therapy is the Reference 25, not 3. This point should be discussed in the introduction.*

A: Thanks for your kind suggestion. After conducting careful literature review and verification, the first-time application of GOx for in vivo cancer starvation therapy is the Reference 26 (Angew. Chem. Int. Ed. 2017, 56, 1229-1233), not 25 or 3. We thus added the discussion about this point in the introduction and justified the cited reference. Please check **page 2** and **reference 2** in the revised manuscript, as marked with YELLOW backgrounds.

Q3. *Since GOx could consume glucose and inhibit ATP production by starvation treatment, the level of intracellular ATP upon above treatments thus should be investigated.*

A: Thanks for your kind suggestion. We investigated the level of intracellular ATP upon above treatments according to your suggestion. In addition, we also measured the levels of H₂O₂ production and added the related discussions, since GOx and GOx-loaded nanosystem could consume glucose, produce H₂O₂ and inhibit ATP production by starvation treatment. Please see **page 13** and **page 14** in the revised manuscript, and **Supplementary Fig. 10** in the

supporting information. All changes were marked out with YELLOW backgrounds.

Q4. Please added the live-dead assay of tumor cells in vitro upon different treatments to comprehensively evaluate tumor killing effect mediated by nanosystem.

A: Thanks for your kind suggestion. We added the live-dead assay of tumor cells in vitro upon different treatments and described the correspondingly results, as marked with YELLOW backgrounds. Please see **page 13** and **page 14** in the revised manuscript, and **Supplementary Fig. 11** in the supporting information.

Q5. In the section of in vivo biosafety study, although the authors investigated the changes of mouse body weight, major organ pathological analysis, and liver function evaluations, more biosafety experiments need to be provided, for example, the biosafety of kidney, blood compatibility, etc.

A: Thanks for your kind suggestion. We performed the biosafety of kidney and blood compatibility assays, including the blood biochemical levels and hematological parameters (hematocrit value (HCT), mean platelet volume (MPV), hemoglobin (HGB), platelets (PLT), mean corpuscular hemoglobin (MCH), red blood cells (RBC), mean corpuscular hemoglobin concentration (MCHC), red cell distribution width (RDW), mean corpuscular volume (MCV) and white blood cell (WBC)), and kidney function indexes (blood urea nitrogen (BUN) and creatinine (CREA)) of the mice after various administrations, and provided the corresponding discussions. Please see **page 20** in the revised manuscript, and **Supplementary Fig. 21** in the supporting information. All changes were marked with YELLOW backgrounds.

Q6. Please add the detailed procedures about tumor re-challenge assay.

A: Thanks for your kind suggestion. We added the detailed procedures and schematic illustration about tumor re-challenge assay according to your suggestion in the revised manuscript. Please see **page 31** and **Figure 10d** in the revised manuscript. All changes were marked with YELLOW backgrounds.

Q7. Some recent works on GOx-instructed synergistic cancer therapy should be cited.

A: Thanks for your kind suggestion. We cited some recent papers regarding GOx-instructed synergistic cancer therapy. Please see **references 2-4** in the revised manuscript, as marked with YELLOW backgrounds.

REVIEWER COMMENTS

Reviewer #1 (Remarks to the Author):

Accept.

Reviewer #2 (Remarks to the Author):

The authors have addressed all my concerns in the revisions. It is now suitable for publication in Nature Communications.

Reviewer #3 (Remarks to the Author):

The authors supplied another relevant tumor model (4T1 tumor cell-bearing Balb/c mouse model) and studied the in vivo immune response level and in vivo antitumor effect of their nanosystem, they observed similar results to that on B16F10-bearing C57 mouse model. Both two tumor model results primarily confirmed that MOF nanoreactor-based nanosystem exhibited good antitumor efficiency with reinforced immune response activation and relieved immune resistance, which provided a common chance for in vivo therapeutic applications. Meanwhile, the authors added MOF and GOx treatments as controls in animal studies. In general, this manuscript has been revised mostly according to the comments and suggestions of the reviewers. I recommend the acceptance of the manuscript for publication in Nature Communications.

Reviewer #4 (Remarks to the Author):

The paper by Dai et al entitled "Multifunctional metal-organic framework-based nanoreactor for starvation/oxidation improved indoleamine 2,3-dioxygenase-blockade tumor immunotherapy" describes the generation of a novel metal organic framework based multifunctional nanoreactor co-loaded with glucose oxidase (GOx) and indoleamine 2,3-dioxygenase (IDO) inhibitor 1-methyltryptophan (1-MT). They show that this new nano drug is able to penetrate tumor cells in a pH dependent manner and release its cargo and affects tumor cells. In addition, they speculate that starvation/oxidation combined IDO-blockade strengthens antitumor immune response and stimulates immune memory. They also speculate that they observe an inhibition of tumor growth and metastasis in vivo.

While the premise of this work is interesting and promising the results do not support the claims of the paper. In addition the tumor immunology studies shown in the paper are a bit limited and would benefit from tumor immunology expertise.

More specifically:

1- The immune characterization shown in multiple figures using flow cytometry is suboptimal and needs expertise from a tumor immunologist. The flow plots that are shown have multiple limitations and are not accurate.

2- The tumor survival and progression are only showing a relative volume. The exact volume should be shown, and the treatment should be started once the tumors are established.

- 3- The breast tumor model is grown on the flank while it should be implanted in the mammary fat pad.
- 4- The immune memory experiments are inconclusive. To test for immune memory mice should be implanted at least 100 days after the last treatment to test immune memory.
- 5- It's unclear if the treatment is influencing tumor invasion or that the observed effect is simply due to an impairment of these cells to proliferate.
- 6- The starvation of the cells of the cells should also in theory affect immune cells. It's unclear how this treatment doesn't affect immune cells negatively.
- 7- The paper needs English proof reading as it was difficult to read.

Responses to Reviewers' Comments

Major changes made based on reviewers' comments:

- (1) We provided the cell gating strategy with assistance of a tumor immunologist, and also adjusted the presentation of immune characterization from “the flow plots” to “the flow zebra” or “the flow contour” based on the suggestions from reviewer #4. Please see **Figure 8a,c** and **Figure 10d,d'** in the revised manuscript, and **Supplementary Fig. 14, Fig. 15, Fig. 17-19**, and **Fig. 26** in the Supplementary Information.
- (2) We performed a long-term immunological memory experiment and discussed the related results based on the suggestions from reviewer #4. Please see **page 20** in the revised manuscript, and **Supplementary Fig. 26** in the Supplementary Information.
- (3) We measured the expression of key proteins closely associated with tumor invasion (E-cadherin, MMP2, MMP9) using western blot assay, and discussed the related results based on the suggestions from reviewer #4. Please see **page 14** in the revised manuscript, and **Supplementary Fig. 13** in the Supplementary Information.
- (4) We provided the exact volume change graph upon various treatments based on the suggestions from reviewer #4. Please see **Figure. 9b,b'** in the Supplementary Information.
- (5) We carefully revised the English writing of the manuscript based on the suggestions from reviewer #4. Each change was marked with “YELLOW” background.

Reviewer #1

C: *Accept.*

A: Thanks for your recommendation of publication.

Reviewer #2

C: *The authors have addressed all my concerns in the revisions. It is now suitable for publication in Nature Communications.*

A: Thanks for your recommendation of publication.

Reviewer #3

C: *The authors supplied another relevant tumor model (4T1 tumor cell-bearing Balb/c mouse model) and studied the in vivo immune response level and in vivo antitumor effect of their nanosystem, they observed similar results to that on B16F10-bearing C57 mouse model. Both two tumor model results primarily confirmed that MOF nanoreactor-based nanosystem exhibited good antitumor efficiency with reinforced immune response activation and relieved immune resistance, which provided a common chance for in vivo therapeutic applications. Meanwhile, the authors added MOF and GOx treatments as controls in animal studies. In general, this manuscript has been revised mostly according to the comments and suggestions of the reviewers. I recommend the acceptance of the manuscript for publication in Nature Communications:*

A: Thanks for your recommendation of publication.

Reviewer #4

C: The paper by Dai et al entitled “Multifunctional metal-organic framework-based nanoreactor for starvation/oxidation improved indoleamine 2,3-dioxygenase-blockade tumor immunotherapy” describes the generation of a novel metal organic framework based multifunctional nanoreactor co-loaded with glucose oxidase (GOx) and indoleamine 2,3-dioxygenase (IDO) inhibitor 1-methyltryptophan (1-MT). They show that this new nano drug is able to penetrate tumor cells in a pH dependent manner and release it’s cargo and affects tumor cells. In addition, they speculate that starvation/oxidation combined IDO-blockade strengthens antitumor immune response and stimulates immune memory. They also speculate that they observe an inhibition of tumor growth and metastasis in vivo.

While the premise of this work is interesting and promising the results do not support the claims of the paper. In addition the tumor immunology studies shown in the paper are a bit limited and would benefit from tumor immunology expertise.

A: Thanks for your criticism with instructive comments. Based on your suggestions, we revised the manuscript one by one shown below.

Q1. *The immune characterization shown in multiple figures using flow cytometry is suboptimal and needs expertise from a tumor immunologist. The flow plots that are shown have multiple limitations and are not accurate.*

A: Thanks for your kind suggestion. We corrected the immune characterization with assistance of a tumor immunologist according to your suggestion. First, the cell gating strategy was respectively provided and referred as follows: we firstly select cells by FSC vs SSC to remove the influence of cell debris, and then remove dead cells by live / dead staining, followed by analyzing the targeted proportion of immune cells via special antibody co-staining approach (**Supplementary Fig. 14**). Second, all immune characterization data shown in multiple figures using flow cytometry were supplemented with cell gating strategy respectively (i.e., **Figure 8a,c vs. Supplementary Fig. 14a**; **Figure 10d,d’ vs. Supplementary Fig. 14f**; **Supplementary Fig. 15 vs. Supplementary Fig. 14b**; **Supplementary Fig. 17 vs. Supplementary Fig. 14c**; **Supplementary Fig. 18 vs. Supplementary Fig. 14d**; **Supplementary Fig. 19 vs. Supplementary Fig. 14e**; **Supplementary Fig. 26a,c vs. Supplementary Fig. 14f**) and provided with corresponding quantitative analysis chart based on flow data in the revised manuscript and Supplementary Information (**Figure 8a,c vs. Figure 8b,d & Supplementary Fig. 16**; **Supplementary Fig. 17-19 vs. Figure 8b,d**; **Figure 10d,d’ vs. Figure 10e,e’**; **Supplementary Fig. 26a,c vs. Supplementary Fig. 26b,d**). Furthermore, all the original data of the immune characterization shown in multiple or single figures using flow cytometry were provided as well. More importantly, we also adjusted the presentation of immune data from “**the flow plots**” to “**the flow zebra**” (**Figure 8a,c**; **Figure 10d, d’**; **Supplementary Fig. 18, Fig. 19, Fig. 26**) or “**the flow contour**” (**Supplementary Fig. 15, Fig. 17**) for the accuracy of the data, according

to your suggestion. Please see **Figure 8a,c** and **Figure 10d,d'** in the revised manuscript, and **Supplementary Fig. 14, Fig. 15, Fig. 17-19** and **Fig. 26** in the Supplementary Information.

Q2. *The tumor survival and progression are only showing a relative volume. The exact volume should be shown, and the treatment should be started once the tumors are established.*

A: Thanks for your kind suggestion. The presentation of tumor volume was modified from the relative volume to the exact volume (**Figure 9 b,b'**). Please see **Figure 9 b,b'** in the revised manuscript.

Q3. *The breast tumor model is grown on the flank while it should be implanted in the mammary fat pad.*

A: Thanks for your kind suggestion. We are very sorry for the misunderstanding caused by the missing description about the establishment of breast tumor model. Notably, 4T1 cells were inoculated subcutaneously into the right flank of female Balb/c mice rather than the mammary fat pad. This is another widely recognized and adopted breast tumor model establishment method proven by previous literature [1-5]. We added the related descriptions about the establishment of breast tumor model in the revised manuscript. Please see **page 28**.

References:

- [1] Wang, Q. Y., Wang, Y. P., Ding, J. J., Wang, C. H., Zhou, X. H., Gao, W. Q., Huang, H. W., Shao, F. & Liu, Z. B. A bioorthogonal system reveals antitumour immune function of pyroptosis. *Nature* **579**, 421-426 (2020).
- [2] Huang, L. P., Li, Y. N., Du, Y. A., Zhang, Y. Y., Wang, X. X., Ding, Y., Yang, X. L., Meng, F. L., Tu, J. S., Luo, L. & Sun, C. M. Mild photothermal therapy potentiates anti-PD-L1 treatment for immunologically cold tumors via an all-in-one and all-in-control strategy. *Nat. Commun.* **10**, 4871 (2019).
- [3] Yaari, Z., da Silva, D., Zinger, A., Goldman, E., Kajal, A., Tshuva, R., Barak, E., Dahan, N., HersHKovitz, D., Goldfeder, M., Roitman, J. S. & Schroeder, A. Theranostic barcoded nanoparticles for personalized cancer medicine. *Nat. Commun.* **7**, 13325 (2016).
- [4] Knox, H.J., Hedhli, J., Kim, T. W., Khalili, K., Dobrucki, L. W. & Chan, J. A bio-reducible *N*-oxide-based probe for photoacoustic imaging of hypoxia. *Nat. Commun.* **8**, 1794 (2017).
- [5] Xu, X. L., Deng, G. J., Sun, Z. H., Luo, Y., Liu, J. K., Yu, X. H., Zhao, Y., Gong, P., Liu, G. Z., Zhang, P. F., Pan, F., Cai, L. T. & Tang, B. Z. A biomimetic aggregation-induced emission photosensitizer with antigen-presenting and hitchhiking function for lipid droplet targeted photodynamic immunotherapy. *Adv. Mater.* **33**, 2102322 (2021).

Q4. *The immune memory experiments are inconclusive. To test for immune memory mice should be implanted at least 100 days after the last treatment to test immune memory.*

A: Thanks for your kind suggestion. Generally, the immune memory experiments are

performed over 8-40 days after the mice are re-implanted with tumor cells, due to the limitation of the survive time of mice treated with various administrations having the tumor metastasis and recurrence [1-6]. Therefore, we additionally performed an immunological memory experiment for 50 days after the last treatment according to your suggestion (the mice treated with control/sample groups (20 per group) were generally died at 18 days after tumor resection, attributing to the inherent tumor metastasis and recurrence. About 6 mice survived at 50 days. Taking the validity and significance of the data (the number of parallel groups is at least more than 6) into account, we ended the experiments at 50 days and measured the corresponding immune memory). We discussed and explained the related results and inserted the corresponding data and figures in the revised manuscript and Supplementary Information. Please see **page 20** in the revised manuscript, and **Supplementary Fig. 26** in the Supplementary Information.

References:

- [1] Huang, L. P., Li, Y. N., Du, Y. A., Zhang, Y. Y., Wang, X. X., Ding, Y., Yang, X. L., Meng, F. L., Tu, J. S., Luo, L. & Sun, C. M. Mild photothermal therapy potentiates anti-PD-L1 treatment for immunologically cold tumors via an all-in-one and all-in-control strategy. *Nat. Commun.* **10**, 4871 (2019).
- [2] Wang, T. T., Wang, D. G., Yu, H. J., Feng, B., Zhou, F. Y., Zhang, H. W., Zhou, L., Jiao, S. & Li, Y. P. A cancer vaccine-mediated postoperative immunotherapy for recurrent and metastatic tumors. *Nat. Commun.* **9**, 1532 (2018).
- [3] Choi, K. J., Zhang, S. N., Choi, I. K., Kim J. S. & Yun, C. O. Strengthening of antitumor immune memory and prevention of thymic atrophy mediated by adenovirus expressing IL-12 and GM-CSF. *Gene Therapy* **19**, 711-723 (2012).
- [4] Feng, Y. J., Wu, J. U., Chen, J., Lin, L., Zhang, S. J., Yang, Z. Y., Sun, P. J., Li, Y. H., Tian, H. Y. & Chen, X. S. Targeting dual gene delivery nanoparticles overcomes immune checkpoint blockade induced adaptive resistance and regulates tumor microenvironment for improved tumor immunotherapy. *Nano Today* **38**, 101194 (2021).
- [5] Chen, Q., Xu, L. G., Liang, C., Wang, C., Peng, R. & Liu, Z. Photothermal therapy with immune-adjuvant nanoparticles together with checkpoint blockade for effective cancer immunotherapy. *Nat. Commun.* **7**, 13193 (2016).
- [6] Fang, H. P., Guo, Z. P., Chen, J., Lin, L., Hu, Y. Y., Li, Y. H., Tian, H. Y. & Chen, X. S. Combination of epigenetic regulation with gene therapy-mediated immune checkpoint blockade induces anti-tumour effects and immune response in vivo. *Nat. Commun.* **12**, 6742 (2021).

Q5. *It's unclear if the treatment is influencing tumor invasion or that the observed effect is simply due to an impairment of these cells to proliferate.*

A: Thanks for your kind suggestion. As you indicated, it's unclear if the treatment is influencing tumor invasion or that the observed effect is due to an impairment of these cells to proliferate,

because the nanosystem could also inhibit tumor survival and kill tumor cells. To further clarify the antitumor invasion mechanism of the nanosystem, we thus investigated the expression of key proteins closely associated with tumor invasion (E-Cadherin, MMP2, MMP9) using western blot assay according to your suggestion (**Supplementary Fig. 13**). We also provided the corresponding discussions. The results confirmed again that the good antitumor invasion effect is actually induced by the nanosystem. Please see **page 14** in the revised manuscript, and **Supplementary Fig. 13** in the Supplementary Information.

Q6. *The starvation of the cells of the cells should also in theory affect immune cells. It's unclear how this treatment doesn't affect immune cells negatively.*

A: Thanks for your kind suggestion. Theoretically, the starvation of the cells may also affect immune cells. However, based on the fact that the tumor microenvironment formed by the regulation of tumor metabolism and the competitive energy demand between tumor cells and immune cells play key roles in tumor immune escape and suppression of immune T cells, it is revealed that inhibiting tumor cell metabolism indeed increases tumor immunotherapy effect and immune checkpoint blockade efficiency [1-5]. Meanwhile, the glucose restriction of tumor cells not only recruits immune cells, but also enhances immune cells functions, especially for CD8⁺ T cell effector [1,6,7]. Furthermore, more treatments combining drugs to regulate tumor cells metabolism/starved tumors (*e.g.* BMS-986205, epacadostat, INCB001158, telaglenastat, metformin) with immunotherapy strategies have been used in clinical trials (Clinical Trials reference: NCT02658890, NCT03493945, NCT04231864, NCT02903914, NCT04265534, NCT03048500, NCT03800602, NCT03994744, *etc.*) [1].

In support of this idea, herein we developed a programmable nanosystem to co-deliver starvation therapy drug (GOx) and immunotherapy drug (1-MT) for boosting antitumor immune response through starvation/oxidation combined IDO-blockade immunotherapy. This nanosystem not only strategically overcomes biobarriers and improves the delivery efficiency via the weakly acidic tumor microenvironment-sensitive size/charge transition, but also effectively reinforces immune response activation with reduced immune resistances.

As for the effect of this nanosystem on immune cells, we firstly constructed an *in vitro* co-culture model composed of B16F10 cells and lymphocytes to examine the proliferation of T cells and evaluate effect of this treatment on immune cells (**Figure. 6c-e**). Compared to the control, the proportion of proliferating T cells significantly increased after the treatment with 1-MT and 1-MT loaded formulations (PCP-Mn-DTA@1-MT and PCP-Mn-DTA@GOx@1-MT), attributing to expression inhibition of IDO and recovery of immune T cells caused by 1-MT from the nanosystem. Notably, **the introduction of starvation therapy drug DOX has not caused significant suppression of immune cell proliferation** (PCP-Mn-DTA@1-MT vs. PCP-Mn-DTA@GOx@1-MT, **Figure. 6c-e**), implying this treatment doesn't affect immune cells negatively. Secondly, we studied the biosafety of the nanosystem and its blood compatibility (**Supplementary Fig. 23**) including the blood biochemical levels and hematological parameters

(e.g. white blood cell (WBC)) that are associated with the activity of immune cells. The results demonstrated that the nanosystem possessed good biocompatibility compared with control, implying that this treatment may not negatively regulate the activity of immune cells. More importantly, we further investigated the various immune cell levels *in vivo* (**Figure 8a-d, Supplementary Fig. 14-19**), which directly associated with its activity. The results suggested that the nanosystem effectively promoted immune cell recruitment and tumor infiltration, evidenced by the recruitment of CTLs, DC cells, B cells and NK cells and the inhibition of Treg cell proliferation, indicating amplified anti-tumor immune response and effectively suppressed immune tolerance. This phenomenon further confirmed that this treatment did not negatively affect immune cell, otherwise, there will be no such obvious tumor immunotherapy effect. In addition, we subsequently monitored the long term immune-memory effect (**Figure 10d,e,d',e', Supplementary Fig. 26**), indicating again the long-lasting effectiveness of immune-memory within 50 days. This assay also suggested that there was no obvious negative effect on immune cell activity upon this treatment. Please see **Figure 6c-e, Figure 8a-d and Figure 10d,e,d',e'** in the revised manuscript, and **Supplementary Fig. 14-19 and Supplementary Fig. 26** in the Supplementary Information.

Reference:

- [1] Cerezo, M. & Rocchi, S. Cancer cell metabolic reprogramming: a keystone for the response to immunotherapy. *Cell Death Dis.* **11**, 964 (2020).
- [2] Elia, I. & Haigis, M. C. Metabolites and the tumour microenvironment: from cellular mechanisms to systemic metabolism. *Nat. Metab.* **3**, 21-32 (2021).
- [3] Li, X. Y., Wenes, M., Romero, P., Huang, S. C. C., Fendt, S. M. & Ho, P. C. Navigating metabolic pathways to enhance antitumour immunity and immunotherapy. *Nat. Rev. Clin. Oncol.* **16**, 425-441 (2019).
- [4] Wei, F., Wang, D., Wei, J. Y., Tang, N. W., Tang, L., Xiong, F., Guo, C., Zhou, M., Li, X. L., Li, G. Y., Xiong, W., Zhang, S. S. & Zeng, Z. Y. Metabolic crosstalk in the tumor microenvironment regulates antitumor immunosuppression and immunotherapy resistance. *Cell Mol. Life Sci.* **78**, 173-193 (2021).
- [5] DePeaux, K., Delgoffe, G.M. Metabolic barriers to cancer immunotherapy. *Nat. Rev. Immunol.* **21**, 785-797 (2021).
- [6] Püschel, F., Favaro, F., Redondo-Pedraza, J., Lucendo, E., Iurlaro, R., Marchetti, S., Majem, B., Eldering, E., Nadal, E., Ricci, J. E., Chevet, E. & Muñoz-Pinedo, C. Starvation and antimetabolic therapy promote cytokine release and recruitment of immune cells. *Proc. Natl. Acad. Sci. U.S.A.* **117**, 9932-9941 (2020).
- [7] Ajona, D., Ortiz-Espinosa, S., Lozano, T., Exposito, F., Calvo, A., Valencia, K., Redrado, M., Remírez, A., Lecanda, F., Alignani, D., Lasarte, J. J., Macaya, I., Senent, Y., Bértolo, C., Sainz, C., Gil-Bazo, I., Eguren-Santamaría, I., Lopez-Picazo, J. M., Gonzalez, A., Perez-Gracia, J. L., de Andrea, C. E., Vicent, S., Sanmamed, M. F., Montuenga, L. M. & Pio, R. Short-term starvation

reduces IGF-1 levels to sensitize lung tumors to PD-1 immune checkpoint blockade. *Nat. Cancer* **1**, 75-85 (2020).

Q7. *The paper needs English proof reading as it was difficult to read.*

A: Thanks for your kind suggestion. We carefully revised the English writing of the manuscript with assistance of a native English speaker.

REVIEWERS' COMMENTS

Reviewer #4 (Remarks to the Author):

The authors have addressed my concerns in this round of revisions. I believe it's suitable for publication.

Responses to Reviewers' Comments

Reviewer #4

The authors have addressed my concerns in this round of revisions. I believe it's suitable for publication.

Answer: Thanks for your recommendation of publication.